# ZAP targets aberrant mRNA transcripts encoding proteins with defective signal peptides for degradation

Akruti Shah[1,3], Aaztli R Coria[1,3], Britnie Santiago Membréno [ID][1,3], Emilien Orgebin[1], Jennifer T Miller[1], Wilfried Guiblet[2] & Colin Chih-Chien Wu [ID][1✉]

## Abstract

The endoplasmic reticulum (ER) is an important site for accurate folding and processing of secretory and membrane proteins. Signal peptides within such proteins are recognized by the signal recognition particle (SRP), which guides them to the ER. When this process is impaired, cells rely on quality control mechanisms to prevent the accumulation of misfolded or mislocalized proteins. One of these mechanisms, known as regulation of aberrant protein production (RAPP), detects nascent proteins with aberrant signal peptides and degrades their mRNA templates. Using functional genetic screens, we identify the zinc finger antiviral protein (ZAP) as a key component of the RAPP pathway. Proteomics and enhanced UV-crosslinking and immunoprecipitation (eCLIP) experiments reveal that the short isoform ZAP-S associates with SRP components and facilitates degradation of aberrant mRNAs. ZAP-S recognizes faulty proteins early in their biogenesis and targets their corresponding mRNAs for degradation. Loss of ZAP activates the unfolded protein response and the downstream integrated stress response, highlighting its central role in safeguarding protein targeting and maintaining cellular homeostasis.

**Keywords** RAPP; GRN; Signal Peptide; ZAP; mRNA Quality Control
**Subject Category** Translation & Protein Quality

## Introduction

Cells dedicate substantial energy to quality control, allowing accurate transmission of genetic information from DNA to mRNA and ultimately to proteins that function across various cellular compartments. Although proteins destined for secretion or delivery to subcellular localizations are synthesized in the cytosol, they undergo maturation and proper folding in the endoplasmic reticulum (ER). The regulation of aberrant protein production (RAPP) pathway is a critical quality control mechanism that preemptively prevents the accumulation of potentially mislocalized proteins, but little is known about it. Specifically, the components mediating RAPP are unclear as is the mechanism underlying this quality control pathway.

Proteins targeted to the ER typically contain an N-terminal signal peptide, characterized by at least one positively charged amino acid followed by a hydrophobic region. The signal recognition particle (SRP) recognizes a signal peptide as it emerges from the ribosome exit tunnel (Walter and Johnson, 1994). The resulting ribosome-nascent chain-SRP complex is then co-translationally targeted to the SRP receptor in the ER membrane. The nascent polypeptide is subsequently translocated to the ER lumen, where the signal peptide is cleaved by signal peptidases (Liaci et al, 2021; Gemmer and Förster, 2020). Properly folded proteins exit the ER and reach their intended cellular destinations, whereas misfolded proteins are targeted for elimination via ER-associated degradation (ERAD), a process involving cytosolic ubiquitination and proteasomal degradation (Walter and Ron, 2011; Karagöz et al, 2019; Christianson and Ye, 2014).

The hydrophobic region of a signal peptide is characterized by its high proportion of hydrophobic amino acids, such as leucine (36.93%), alanine (9.68%), and valine (8.8%). In contrast, hydrophilic residues like tyrosine (0.7%), asparagine (0.39%), glutamine (1.46%), and lysine (0.41%) are strongly disfavored (Gutierrez Guarnizo et al, 2023). This hydrophobic core is crucial for SRP recognition as it facilitates the insertion of a signal peptide into the binding pocket of SRP54 within the SRP complex (Janda et al, 2010). Disruption of this core compromises SRP recognition and may result in defective targeting of nascent polypeptides.

The initial interaction between the SRP and a signal peptide is crucial for both accurate protein targeting and the stability of the corresponding mRNA. Mutations that reduce the hydrophobicity of a signal peptide impair SRP recognition, allowing AGO2 to bind the faulty signal peptide and target the mutant mRNA for degradation. This quality control process, known as regulation of aberrant protein production (RAPP), preemptively prevents the synthesis of potentially mislocalized proteins (Karamyshev et al, 2014). Mutations in the signal peptides of preprolactin (PPL) and progranulin (GRN) mRNAs have been shown to trigger the RAPP pathway (Karamyshev et al, 2014; Pinarbasi et al, 2018). RAPP is

[1]Center for Cancer Research, National Cancer Institute, National Institutes of Health, Frederick, MD 21702, USA. [2]Advanced Biomedical Computational Science, Frederick National Laboratory for Cancer Research, Frederick, MD 21702, USA. [3]These authors contributed equally: Akruti Shah, Aaztli R Coria, Britnie Santiago Membréno.
✉E-mail: colin.wu2@nih.gov

thought to contribute to progranulin haploinsufficiency (Pinarbasi et al, 2018), a major cause of frontotemporal lobar degeneration (FTLD) (Gass et al, 2006; Baker et al, 2006). Notably, while AGO2 is required for the decay of mutant PPL mRNA, it is dispensable for the elimination of mutant GRN mRNA. These findings indicate the molecular mechanisms triggered by aberrant signal peptides are not yet fully understood.

Beyond the RAPP pathway, AGO2 has been shown to degrade ER-targeted mRNAs that encode a subset of secretory proteins via the RNA-induced silencing complex (RISC)-mediated mRNA decay machinery in a process termed ER-associated RNA silencing (ERAS) (Efstathiou et al, 2022). The slicer activity of AGO2 is dispensable for the RAPP pathway, and yet it is critical for the ERAS pathway (Efstathiou et al, 2022; Karamyshev et al, 2014). While the processes governing ERAD are well-characterized, the RAPP pathway remains poorly defined. The ambiguous function of AGO2 in ER quality control adds an additional challenge, leaving the RAPP pathway largely unexplored.

The ER serves as a central hub for protein synthesis and quality control, utilizing multiple mechanisms to prevent the potentially toxic accumulation of misfolded proteins (Krshnan et al, 2022). Accumulation of misfolded proteins in the ER activates the unfolded protein response (UPR), an adaptive pathway that restores cellular homeostasis (adaptive UPR) (Friedlander et al, 2000; Travers et al, 2000). UPR activation reduces the burden of unfolded proteins through ERAD and selectively degrades ER-bound mRNAs via regulated IRE1-dependent mRNA decay (RIDD). However, if these pathways fail to eliminate unfolded proteins, the terminal UPR is initiated, leading to apoptotic cell death. This process involves phosphorylation of eukaryotic initiation factor 2α (eIF2α) and the subsequent translational upregulation of ATF4, a hallmark of the integrated stress response (ISR), as well as ATF5 and CHOP, two critical downstream pro-apoptotic transcription factors. ATF4 further activates ATF5 and CHOP transcription, initiating an apoptotic gene expression program (Teske et al, 2013; Wek, 2018).

In this study, we present functional genetic screens that identify ZAP as the primary RNA-binding protein that participates in the RAPP quality control pathway. ZAP, also known as PARP13 and ZC3HAV1, was initially identified as a zinc finger antiviral protein (Todorova et al, 2015; Gao et al, 2002). It binds viral and non-viral RNAs to direct them for degradation via the cellular mRNA decay machinery. Using proteomic and enhanced UV crosslinking and immunoprecipitation (eCLIP) experiments, we demonstrate that the ZAP small isoform (ZAP-S) associates with components of the SRP complex and that this interaction is critical for the RAPP-mediated mRNA degradation. Moreover, we show that loss of ZAP activates the UPR and the downstream ISR. Together, we propose a surveillance model wherein ZAP facilitates the degradation of mRNAs encoding aberrant signal peptides, thereby preventing the accumulation of potentially toxic misfolded and mislocalized proteins.

# Results

## Establishing a reporter system for studying RAPP

To uncover the molecular mechanism underlying RAPP, we developed an inducible dual-fluorescence reporter consisting of the bovine preprolactin (PPL) signal peptide in frame with mCherry followed by EGFP expressed from a separate promoter as an internal control (Fig. 1A). Two (Δ2 L) or four (Δ4 L) leucines in the hydrophobic domain of the PPL signal peptide were deleted to disrupt its interaction with SRP and render it a RAPP substrate (Fig. 1A) as reported previously (Karamyshev et al, 2014). As expected, RT-qPCR experiments in human K562 (lymphoblastoid leukemia) and HeLa (cervical carcinoma) cells transduced with the PPL reporters revealed a 2- to 2.5-fold reduction in PPL-Δ2 L mRNA levels compared to the WT reporter (Figs. 1B and EV1A). Notably, the PPL-Δ4 L deletion showed only a modest decrease in mRNA levels relative to WT (Figs. 1B and EV1A).

We next assessed reporter levels by immunoblotting and observed a significant reduction in mature mCherry protein levels in cells expressing PPL-Δ2 L (Figs. 1C and EV1B, lanes 1–2). Similar to the RT-qPCR results, the PPL-Δ4 L signal peptide resulted in a more modest reduction in protein levels, and yet immunoblotting revealed two distinct mCherry bands (Figs. 1C and EV1B, lane 3). We reasoned that the upper species likely represents the reporter protein with an aberrant uncleaved PPL signal peptide (referred to as SP*-mCherry). This is likely due to reduced SRP engagement together with inefficient cleavage of the signal peptide (also see Appendix Fig. S1A), as previously reported for mutant preproparathyroid hormone (Karaplis et al, 1995). Consistent with the secretory role of the PPL signal peptide, the PPL-WT reporter was more efficiently secreted into the culture media compared to the Δ2 L and Δ4 L reporters (Fig. 1C). In contrast to its cleaved form, SP*-mCherry species was not detected in the media (Fig. 1C, lane 3).

We next determined the subcellular localization of the reporters by immunofluorescence microscopy and observed a high degree of co-localization between the PPL-Δ2 L reporter and the ER, compared to the cytosolic EGFP control (Figs. 1D,E and EV1C). In contrast, SP*-mCherry species derived from the PPL-Δ4 L reporter showed a lower degree of co-localization with the ER (Appendix Fig. S1B,C). To further characterize RAPP, we chose to focus on the PPL-Δ2 L reporter, which produces a single protein product.

## RAPP reduces ribosome loading of target mRNAs

While RAPP reporters showed differences in mRNA levels and concomitant protein production, we next asked whether there was a difference in the extent of translation. We generated stable K562 cell lines that express either the PPL-WT or Δ2 L reporter, enabling quantitative measurement of protein abundance by flow cytometry (Fig. EV1D). K562 cells were selected for subsequent analysis owing to their high responsiveness to ER stress and amenable to systematic genome perturbation (Piccolis et al, 2019). Notably, the reduction in protein abundance ($0.11 \pm 0.02$-fold by flow cytometry) was greater than that in mRNA abundance ($0.29 \pm 0.04$-fold by RT-qPCR) (Fig. 1F); we note that these calculations did not account for secreted reporter proteins. This result suggests that the translation efficiency (TE)—defined as the ratio of proteins produced per mRNA molecule—of the PPL-Δ2L reporter is repressed (Fig. 1F). To further investigate this observation, we performed RNA sequencing (RNA-seq) on total RNA and ribosome profiling (Ribo-seq) to quantify the mRNA abundance and ribosome occupancy of the PPL-WT and Δ2 L reporters, respectively. We calculated ribosome occupancy per

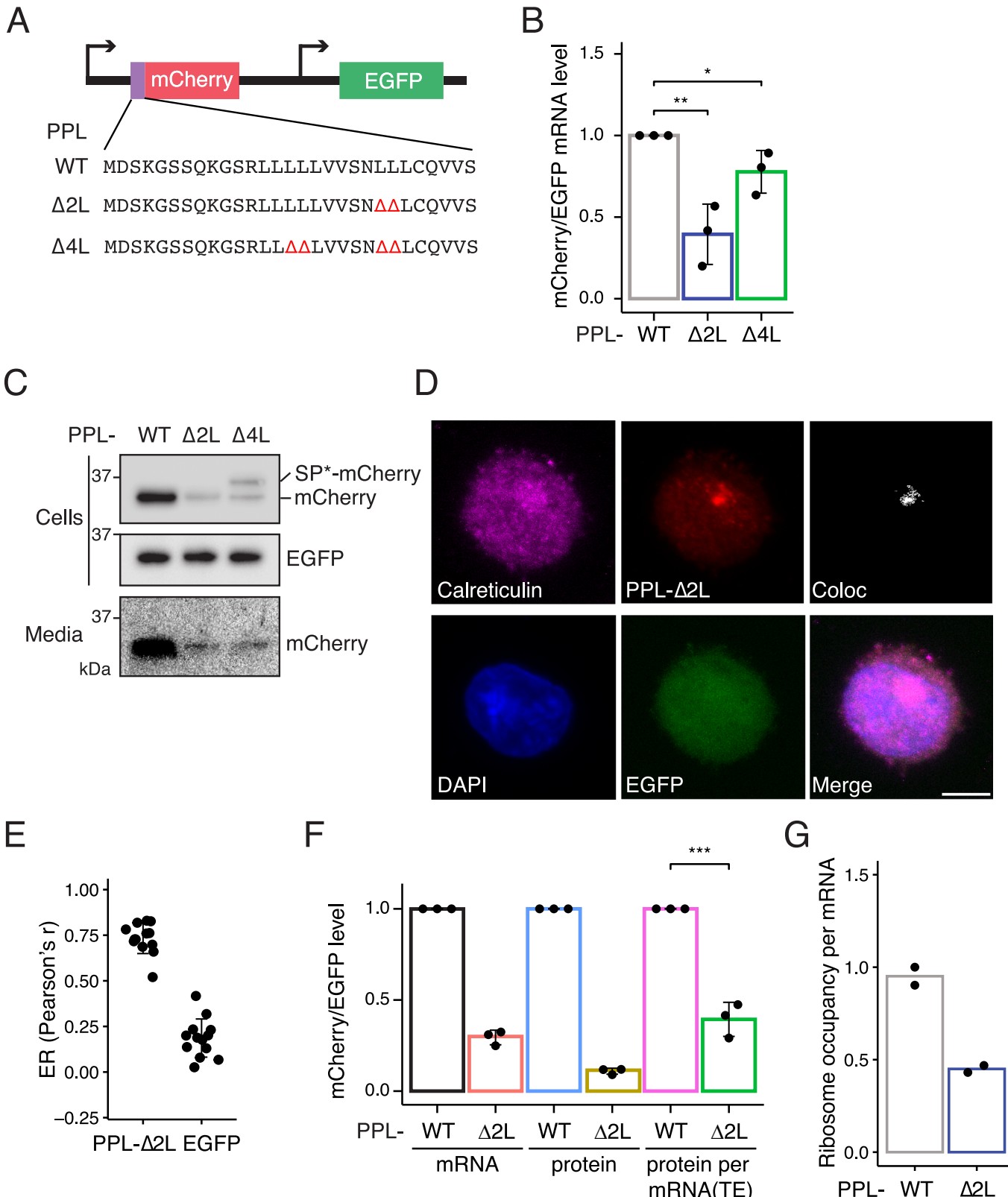

◀ **Figure 1. A reporter for studying RAPP.**

(A) Dual fluorescence RAPP reporter used in this study. Wild-type or mutant preprolactin (PPL) signal peptide sequences are fused to mCherry. Leucine deletions that reduce PPL hydrophobicity are indicated in red. Expression of mCherry is inducible by doxycycline treatment, whereas EGFP is constitutively expressed and used for normalization. (B) Relative mCherry mRNA levels determined by RT-qPCR normalized to EGFP mRNA levels in K562 cells transduced with the PPL-WT, Δ2 L, or Δ4 L reporter. The bar plot represents the mean and error bars indicate standard deviations ($n = 3$). Student's $t$ test is indicated by asterisks. *$P < 0.05$; ***$P < 0.001$. (C) Representative immunoblots ($n = 3$) of mCherry and EGFP expression in cells and culture media. EGFP serves as an expression and loading control. SP*-mCherry denotes the reporter species retaining an uncleaved signal peptide. (D) Representative images of cells stained with anti-Calreticulin, anti-FLAG (PPL-Δ2 L reporter), and anti-GFP antibodies. Colocalization of calreticulin and the PPL-Δ2 L reporter is shown. Scale bar, 5 μm. (E) Quantification of PPL-Δ2 L reporter or EGFP colocalization with the calreticulin-positive pixels using Pearson's r values and error bars denote standard deviations ($n = 13$). Also see Fig. EV1C. (F) Quantification of the PPL-WT and Δ2 L reporters at mRNA and protein levels, measured by RT-qPCR and flow cytometry (median of mCherry/EGFP distribution), respectively. Translation efficiency (protein per transcript) was calculated as the ratio of protein to mRNA abundance for each reporter. Error bars denote standard deviations ($n = 3$). Student's $t$ test is indicated by asterisks. ***$P < 0.001$. (G) Ribosome occupancy per mRNA was calculated by normalizing ribosome footprints density to mRNA abundance for the PPL-WT and Δ2 L reporters (biological replicates, $n = 2$). Source data are available online for this figure.

mRNA by normalizing ribosome footprints to RNA-seq data for the reporters and observed a ~2.5-fold reduction for the PPL-Δ2 L reporter (Fig. 1G), indicating that the Δ2 L reporter transcripts are translated by fewer ribosomes. Together, these results indicate that RAPP mitigates aberrant protein production by both repressing target mRNA translation and promoting transcript degradation.

## Canonical RIDD and RQC factors are not required for RAPP

To gain mechanistic insights into how cells deal with secretory proteins bearing an aberrant signal peptide, we next sought to identify genes required for the pathway. Loss of genes linked to RAPP should increase the mCherry/EGFP ratio. To assess the suitability of the RAPP reporter for Cas9-mediated gene editing, we first tested a single guide RNA (sgRNA) targeting mCherry via lentiviral transduction. Ablation of mCherry resulted in a drastic decrease in the mCherry signal as expected, with a minimal impact on EGFP levels (Fig. EV1E). The ribonucleoprotein SRP recognizes the signal peptide through its component SRP54 (Zopf et al, 1990; Römisch et al, 1990; Janda et al, 2010). Given that a recent study has shown that knockdown of SRP54 destabilizes mRNAs encoding secretory proteins (Tikhonova et al, 2022), we tested its impact on the RAPP reporters. Indeed, ablation of SRP54 by sgRNAs reduced both PPL-WT and Δ2 L reporter levels (Fig. 2A). Taken together, these data demonstrate the robustness of our RAPP reporter system and its potential for identifying regulators of the quality control pathway.

Upon activation of UPR, IRE1 (ERN1 in accordance with human nomenclature) initiates regulated IRE1-dependent mRNA decay (RIDD) to selectively degrade ER-bound mRNA (Hollien and Weissman, 2006). This pathway also requires PELO and HBS1L, which are implicated in ribosome rescue, mRNA no-go decay (NGD), and RQC (Shoemaker et al, 2010; Guydosh et al, 2017; Guydosh and Green, 2014). To test whether these RIDD factors participate in the RAPP pathway, we employed CRISPR/Cas9 to ablate them. Flow cytometric analyses showed no stabilization of the PPL-Δ2 L reporter upon the depletion of these genes using two independent sgRNAs (Fig. EV2A,B,D). In addition, we also assessed the putative NGD and RQC endoribonuclease N4BP2, the human homolog of yeast Cue2 (D'Orazio et al, 2019) and *C. elegans* NONU-1 (Glover et al, 2020). Ablation of N4BP2 did not change protein expression from the PPL-Δ2 L reporter (Fig. EV2C,D).

## A functional genetic screen identifies ZAP as a RAPP regulator

In order to identify genes that promote RAPP, we designed a genome-wide CRISPR/Cas9 screen based on fluorescence-activated cell sorting (FACS) of PPL-Δ2L-expressing cells. We transduced both PPL-WT and PPL-Δ2 L reporter cell lines with the Brunello library, which targets 19,114 known human protein-coding genes (Shalem et al, 2014) at a low multiplicity of infection. This resulted in a pool of single-gene knockouts. Ablation of genes that promote RAPP is expected to increase the mCherry/EGFP ratio. Therefore, we used FACS to isolate the top 15% of transduced cells with single-gene knockouts that resulted in the highest mCherry/EGFP signal ratios (Fig. 2B). We then deep-sequenced sgRNAs from these sorted cells. To calculate enrichment of sgRNAs, we compared the isolated cells to the unsorted population. Argonaute 2 (AGO2), previously shown to bind the PPL signal peptide to promote RAPP activity (Karamyshev et al, 2014), emerged as a top hit in our PPL-Δ2 L reporter cells, validating our genetic screen. Two other genes, ZAP and STK11, also stood out as significant hits with high confidence based on a stringent statistical cutoff (Fig. 2C; Dataset EV1). Importantly, none of these hits were significantly enriched in the screen carried out with PPL-WT-expressing cells (Fig. EV2E).

To validate the screen results, individual top-scoring sgRNAs were cloned and transduced into PPL-WT and Δ2 L reporter cells. Ablation of AGO2 and ZAP led to an increased mCherry/EGFP ratio in PPL-Δ2 L reporter cells but not in WT counterparts (Fig. 2D–F). Ablation of STK11 had little effect on stabilizing mCherry levels (Fig. EV2F). Given that depletion of SRP components induces RAPP (Tikhonova et al, 2022; Karamyshev et al, 2014), we next asked whether ZAP contributes to this process. To this end, we depleted SRP54 in ZAP KO cells and observed reduced RAPP activity (Fig. EV2G), supporting the notion that ZAP promotes RAPP activation under SRP-deficient conditions. Together, these data confirmed the reported role of AGO2 in regulating RAPP and prompted further investigation into the function of ZAP in the mRNA quality control pathway.

ZAP, also known as ZC3HAV1 (Zinc Finger CCCH-type, antiviral 1) and PARP13, has previously been implicated in the innate immune response, where it inhibits viral RNA replication (Todorova et al, 2015). To further investigate the role of ZAP in RAPP, we first generated ZAP knockout (KO) cell lines (Fig. 3A). As expected, PPL-Δ2 L mRNA and protein levels were markedly

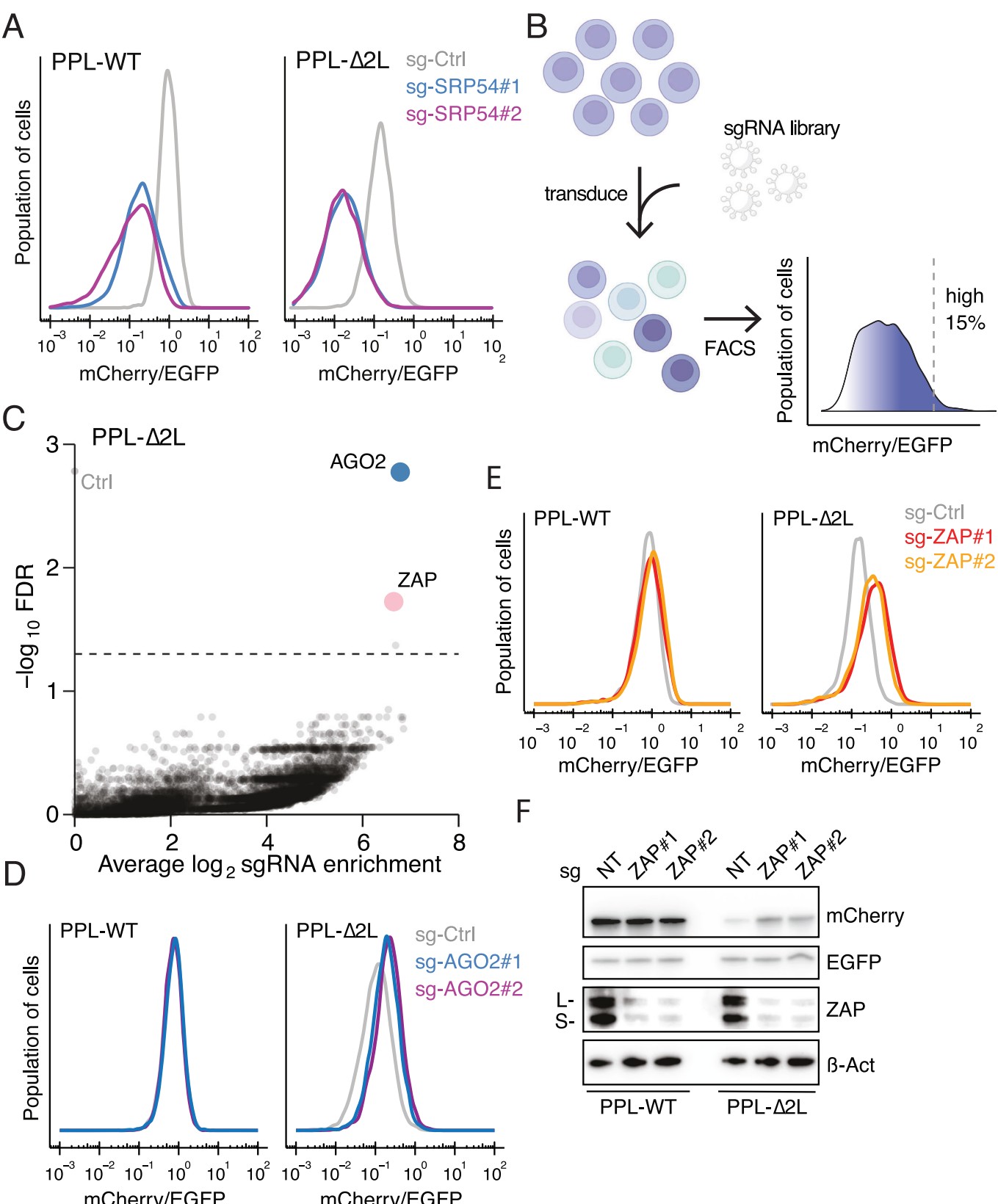

**Figure 2. CRISPR screens identify RAPP regulators.**

(A) Flow cytometric analysis of cells expressing PPL-WT (left) and PPL-Δ2 L reporter (right) after ablation of SRP54 by two independent sgRNAs. (B) Schematic of functional genetic screens performed using the RAPP reporters. Cells expressing PPL-WT or PPL-Δ2 L reporter were lentivirally transduced. After puromycin selection, cells were sorted to obtain the top 15% brightest mCherry signal (normalized to EGFP signal). (C) Volcano plot showing effect size (average log$_2$ fold change, x axis) versus FDR (-log(FDR), y axis) with significantly enriched genes labeled (FDR ≤ 0.05). Ctrl, control sgRNAs. (D, E) Representative flow cytometric analysis of cells expressing the PPL-WT (left) or PPL-Δ2 L reporter (right) after sgRNA-mediated ablation of AGO2 (D) and ZAP (E) using top two independent sgRNAs identified from (C). (F) Immunoblots of mCherry and EGFP expression levels upon sgRNA-mediated depletion of ZAP in cells expressing the PPL-WT or PPL-Δ2 L reporter. β-actin serves as a loading control. ZAP isoforms are indicated. Source data are available online for this figure.

stabilized in two isogenic ZAP KO cell lines, as measured by RT-qPCR and flow cytometry (Fig. 3A–C). Previous work indicates that ZAP mediates translational repression by sequestering the translation initiation factor eIF4A; this repression is required for viral mRNA degradation (Zhu et al, 2012; Todorova et al, 2015). To assess whether ZAP is responsible for the translational repression of the PPL-Δ2 L reporter (Fig. 1G). We next performed RNA-seq and Ribo-seq with ZAP KO cells. While RNA-seq results confirmed stabilization of the PPL-Δ2 L reporter mRNA (Dataset EV2), ZAP KO did not increase ribosome occupancy per mRNA on the PPL-Δ2 L reporter (Fig. 3D; Appendix Fig. S2). These findings argue that ZAP promotes RAPP predominantly through mRNA degradation, and that loss of ZAP decouples mRNA degradation from translational repression.

AGO2 has been shown to promote mRNA degradation during RAPP (Karamyshev et al, 2014) and miRNA-mediated translational repression (Bartel, 2018). Given that ZAP had no role in translational repression during RAPP (Fig. 3D), we next tested whether AGO2 could repress the PPL-Δ2 L reporter translationally. To do so, we performed an epistasis experiment by depleting AGO2 in ZAP KO cells where the PPL-Δ2 L reporter was still translationally repressed. RT-qPCR confirmed that AGO2 knockdown stabilizes PPL-Δ2 L mRNA (Fig. EV3A). However, RT-qPCR and flow cytometric analyses showed that depletion of AGO2 in ZAP KO cells did not further stabilize the reporter at either mRNA and protein level (Fig. EV3B,C), suggesting that AGO2 does not translationally repress the PPL-Δ2 L reporter. Taken together, these results demonstrate a functional epistatic relationship between ZAP and AGO2 in promoting RAPP.

## The ZAP-S isoform is critical for mRNA degradation during RAPP

ZAP has two major isoforms, small (ZAP-S) and large (ZAP-L) (Fig. 3E), generated by alternative splicing and intronic polyadenylation (Ly et al, 2022; Li et al, 2019; Schwerk et al, 2019). While both ZAP isoforms play a role in coordinating the innate antiviral immune response and modulating the UPR (Schwerk et al, 2019; Ly et al, 2022), it remains unknown which isoform contributes to the quality control pathway. To address this, we transduced isoform-specific sgRNAs to ablate either ZAP-S or ZAP-L in PPL-Δ2 L reporter cells. Specifically, a sgRNA pair targeting the first L isoform-specific exon was co-transduced to ablate ZAP-L. In addition, we used a distinct sgRNA pair to remove the intronic polyadenylation signal and ablate ZAP-S generation while preserving ZAP-L (Fig. EV3D). Immunoblotting confirmed specific depletion of ZAP isoforms and flow cytometric analysis revealed that ZAP-S ablation strongly stabilized mCherry levels

whereas depletion of ZAP-L did not (Fig. 3F). Furthermore, ZAP-S KO clonal lines showed stably elevated PPL-Δ2 L reporter at protein and mRNA levels (Figs. 3G,H and EV3E).

Progranulin (GRN), a secreted immunity factor, is another established substrate of the RAPP pathway. Pathogenic mutations within the GRN signal peptide are associated with frontotemporal lobar degeneration (FTLD) (Gass et al, 2006; Baker et al, 2006) and trigger the RAPP-mediated mRNA decay independently of AGO2 (Pinarbasi et al, 2018). To test whether ZAP-S facilitates the decay of mutant GRN mRNA, in addition to regulating the PPL-Δ2 L reporter, we generated reporter cell lines using either the wild-type GRN signal peptide or its pathogenic A9D counterpart (alanine to aspartate mutation) (Fig. 3I). Knockdown of ZAP-S using isoform-specific siRNA stabilized the GRN-A9D reporter mRNA (Fig. 3J), indicating that ZAP contributes to the selective degradation of mutant GRN mRNA.

## ZAP-S associates with the SRP and SRP receptor

Previous work showed that ZAP restricts HIV-1 replication by interacting with the cellular endoribonuclease, N4BP1 and its paralog KHNYN (Ficarelli et al, 2019; OhAinle et al, 2018; Anantharaman and Aravind, 2006). In addition, ZAP, N4BP1, and KHNYN are key components of the TRIM25-mediated RNA surveillance pathway for unmodified exogenous mRNA (Kim et al, 2025). To test whether ZAP promotes RAPP through its interaction with these antiviral endoribonucleases, we ablated these factors using CRISPR/Cas9 and measured the expression levels of the PPL-Δ2 L reporter. Ablation of TRIM25, N4BP1, and KHNYN did not stabilize PPL-Δ2 L reporter expression (Fig. EV3F–H), suggesting that ZAP participates in RAPP independently of its antiviral and exogenous mRNA-sensing functions.

To further explore potential molecular mechanisms by which ZAP-S promotes RAPP, we sought to identify ZAP-S binding partners that might facilitate RAPP. We complemented HA-tagged ZAP-S in ZAP-S KO cells, and immunoprecipitated ZAP-S followed by quantitative mass spectrometry to identify its binding partners (Fig. EV4A). Since ZAP is an RNA-binding protein, we treated cellular lysates with ribonuclease to minimize background due to indirect interactions. Proteomics analysis revealed that ZAP-S binds several LSM proteins (LSM4, 6, 7, and 8) and exosome components (EXOSC4 and EXOSC10) (Fig. 4A; Dataset EV3), consistent with its reported role in degrading viral mRNA by recruiting the exosome (Guo et al, 2007). Gene Ontology (GO) analysis revealed enrichment of genes implicated in protein export and lysosomal function (Fig. EV4B). Notably, SRP72 and SRP68, the SRP receptor proteins (SRPRα and SRPRβ), and an ER luminal chaperone BiP appeared among the enriched proteins under the GO term protein export although their enrichment was less

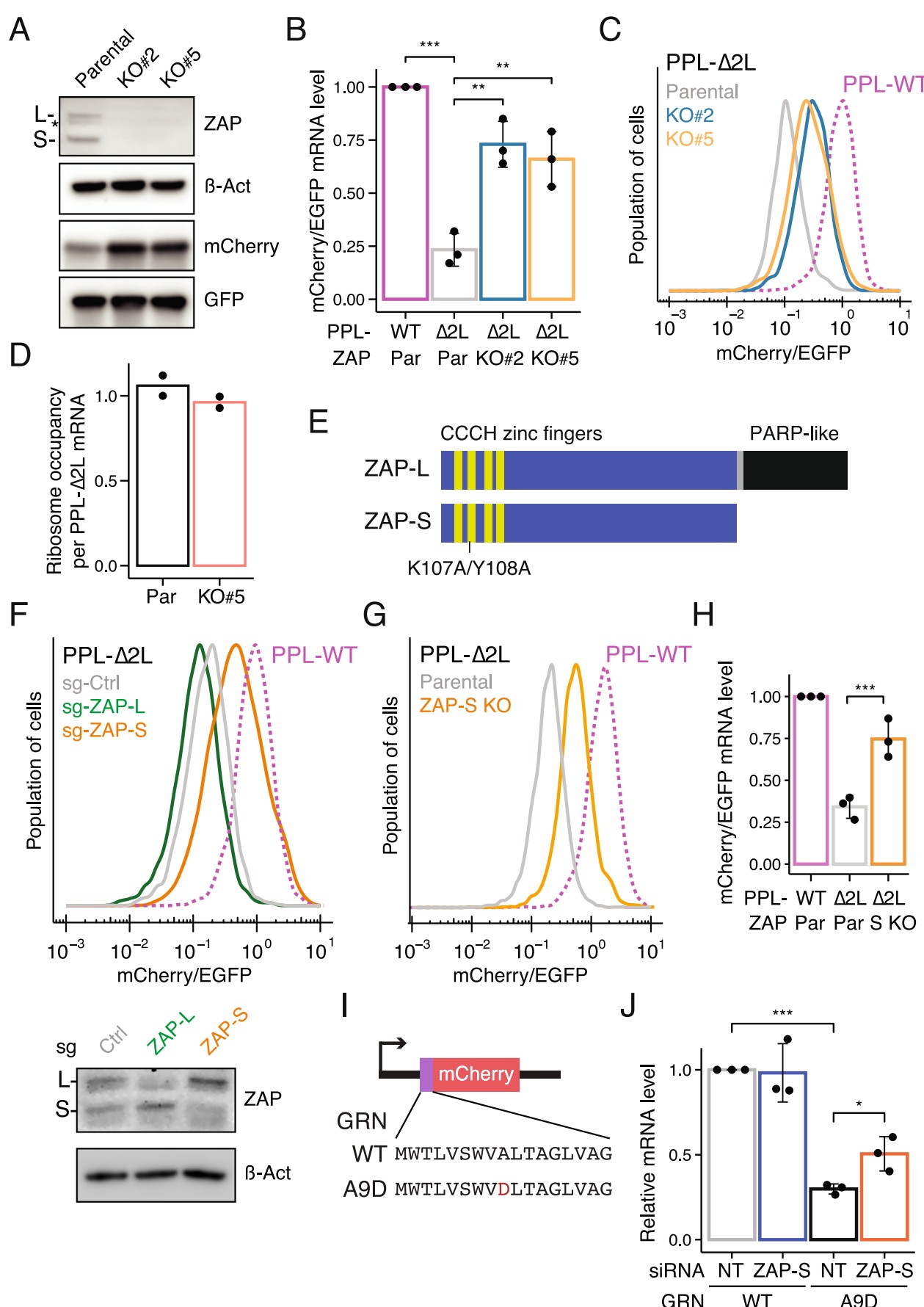

**Figure 3.  ZAP-S participates in RAPP-mediated mRNA decay but not translational repression.**

(A) Immunoblots for ZAP expression in parental and two isogenic ZAP knockout monoclonal lines. Two major isoforms (ZAP-S and ZAP-L) are indicated. β-actin serves as a loading control. (B) RT-qPCR data showing the relative PPL-Δ2 L reporter mRNA levels (normalized to EGFP mRNAs) in parental and two ZAP KO clonal lines compared to PPL-WT reporter mRNA levels in parental cells. Error bars indicate standard deviations ($n = 3$). Student's $t$ test is indicated by asterisks. **$P < 0.01$; ***$P < 0.001$. (C) Representative flow cytometric analysis of mCherry/EGFP ratio in parental and two ZAP KO clonal lines expressing the PPL-Δ2 L RAPP reporter, compared to parental cells expressing the PPL-WT reporter. Representative results from three independent experiments are shown. (D) Ribosome occupancy per mRNA for the PPL-Δ2 L reporter in parental and ZAP KO cells, as determined by normalizing ribosome footprint density to mRNA abundance ($n = 2$). (E) Schematic for the domain organization of ZAP-S and ZAP-L isoforms. Four CCCH zinc fingers are shown in yellow, and a poly (ADP-ribosome) polymerase (PARP)-like domain is shown in black. Mutations (K107A/Y108A) in the second CCCH zinc finger are indicated. (F) Representative flow cytometric analysis of cells expressing the PPL-Δ2 L reporter after sgRNA-mediated ablation of ZAP-S or ZAP-L compared to a control sgRNA (top). Immunoblotting for ZAP upon isoform-specific depletion (bottom). (G) Flow cytometric analysis of mCherry/EGFP ratio in parental and a ZAP-S KO clone expressing the PPL-Δ2 L reporter, compared to parental cells expressing the PPL-WT reporter. (H) RT-qPCR results of relative PPL-Δ2 L reporter mRNA levels (normalized to EGFP mRNAs) in parental, and a ZAP-S KO clonal line. Error bars denote standard deviations ($n = 3$). Student's $t$ test is indicated by asterisks. ***$P < 0.001$. (I) Schematic of human progranulin (GRN) reporter. Signal peptide of progranulin is shown and the pathogenic mutation (A9D) is indicated in red. (J) RT-qPCR results of relative WT and A9D GRN reporter mRNA levels upon siRNA-mediated ZAP-S knockdown. Error bars indicate standard deviations ($n = 3$). Student's $t$ test is indicated by asterisks. *$P < 0.05$; ***$P < 0.001$. Source data are available online for this figure.

pronounced. We first confirmed the interaction between ZAP-S and the SRP by immunoblotting and observed that ZAP-S-SRP54 association was enhanced upon induction of the PPL-Δ2 L reporter (Fig. EV4C). Given that ZAP-S interacts with the SRP and its receptor, we next examined whether it associates with the ER. Biochemical isolation of the ER fraction (microsomes) from cells expressing the PPL-Δ2 L reporter confirmed co-purification of ZAP-S with the ER protein calnexin and SRP receptor component SRPRβ (Fig. 4B). ZAP-S association with the ER likely underlies its interaction with BiP. Moreover, immunofluorescence microscopy confirmed ZAP-S co-localization with calreticulin (Fig. EV4D,E). We note that previous studies reported ZAP-S to be predominantly cytoplasmic whereas ZAP-L localizes to cytoplasmic membranes (Charron et al, 2013; Schwerk et al, 2019; Todorova et al, 2014), suggesting that ZAP-S association with the ER may occur in specific cellular contexts.

Since the SRP is composed of six proteins (SRP9, SRP14, SRP19, SRP54, SRP68, and SRP72) and one RNA component— 7SL RNA, we next examined whether ZAP-S binds 7SL RNA directly. To this end, we performed enhanced UV crosslinking and immunoprecipitation (eCLIP) using cells expressing HA-tagged ZAP-S together with the PPL-Δ2 L reporter. Crosslinked RNA was recovered after immunoprecipitation, converted to small RNA cDNA libraries, and deep sequenced. A total of 1233 reproducible crosslink sites were identified across the transcriptome, spanning both coding and noncoding RNAs (Fig. EV4F,G). Notably, we observed a ~ 5-fold enrichment of ZAP-S binding to helices 6 and 8 of 7SL RNA compared to input samples (Fig. 4C). While most regions of 7SL RNA are bound by SRP proteins (Kobayashi et al, 2018), the identified ZAP-S binding site is not occluded by SRP protein binding (Fig. 4D).

To further validate our findings, we performed electrophoretic mobility shift assays (EMSA) using purified recombinant ZAP-S (Fig. EV4H) and an RNA probe corresponding to the identified binding site (7SL RNA probe). Indeed, we found a significant interaction between ZAP-S and 7SL RNA probe (Figs. 4E and EV4I), with an affinity comparable to that observed for GNCG motif-containing RNAs (Luo et al, 2020). In addition, adding unlabeled 7SL RNA probe competitively displaced ZAP-S RNA binding (Fig. EV4J). Taken together, four orthogonal approaches demonstrate that ZAP-S interacts with the SRP, specifically 7SL RNA, thereby engaging the SRP and associating with the ER.

A previous study showed that ZAP binds CpG-rich RNA specifically via its CCCH zinc finger RNA-binding domains (Luo

et al, 2020). Consistently, we found that mutations in the second CCCH zinc finger RNA binding domain of ZAP-S (K107A/Y108A) significantly abrogated its ability to bind the RNA probe in vitro (Fig. 4F). We next asked whether the CCCH zinc finger mutations abolish the ability of ZAP-S to establish RAPP. Indeed, complementing ZAP-S KO cells with ZAP-S K107A/Y108A mutant showed reduced RAPP activity compared to ZAP-S WT (Figs. 4G and EV4L). Importantly, we did not observe an association between ZAP-S and the PPL-Δ2 L reporter mRNA (Fig. EV4K), suggesting that ZAP-S likely facilitates the RAPP-mediated mRNA decay through its interaction with the SRP rather than direct binding to RAPP substrate mRNAs. Taken together, our findings identify a specific interaction between ZAP-S and the SRP through 7SL RNA and reveal that this interaction is critical for promoting mRNA decay.

## Loss of ZAP induces the unfolded protein response

Both ZAP isoforms have previously been implicated in the UPR and antiviral innate immune signaling (Busa et al, 2024; Ly et al, 2022; Schwerk et al, 2019). A close examination of our RNA-seq results confirmed transcriptional down-regulation of several interferon response genes in ZAP KO cells (Fig. 5A; Dataset EV2), suggesting a desensitized immune state under basal conditions.

With respect to UPR activation, we observed significant upregulation of several canonical UPR hallmark genes (Teske et al, 2013), including ATF5, DDIT3 (CHOP), and ATF6 in ZAP KO cells (Figs. 5A and EV5A). Consistently, several CHOP-dependent genes (Teske et al, 2013) were modestly up-regulated (Fig. EV5B), suggestive of enhanced apoptotic signaling upon ZAP depletion. Furthermore, spliced XBP1 mRNA levels were elevated (Fig. EV5C), and PERK phosphorylation was also modestly elevated in ZAP KO cells (Fig. 5B). As a result, eIF2α phosphorylation was increased in ZAP KO compared to parental cells (Fig. 5C,D), allowing upstream open reading frame (uORF) bypass (Zhou et al, 2008) and translation of the ATF5 coding region (Fig. EV5D). Strikingly, overexpression of ZAP-S, but not ZAP-L, attenuates eIF2α phosphorylation (Fig. 5C,D). Together, these observations suggest that all three ER-resident sensors, ATF6, PERK, and ERN1, are activated in the absence of ZAP. These findings align with the notion that loss-of-function of ZAP triggers the UPR and that ZAP-S isoform restrains UPR activation via RAPP-mediated quality control.

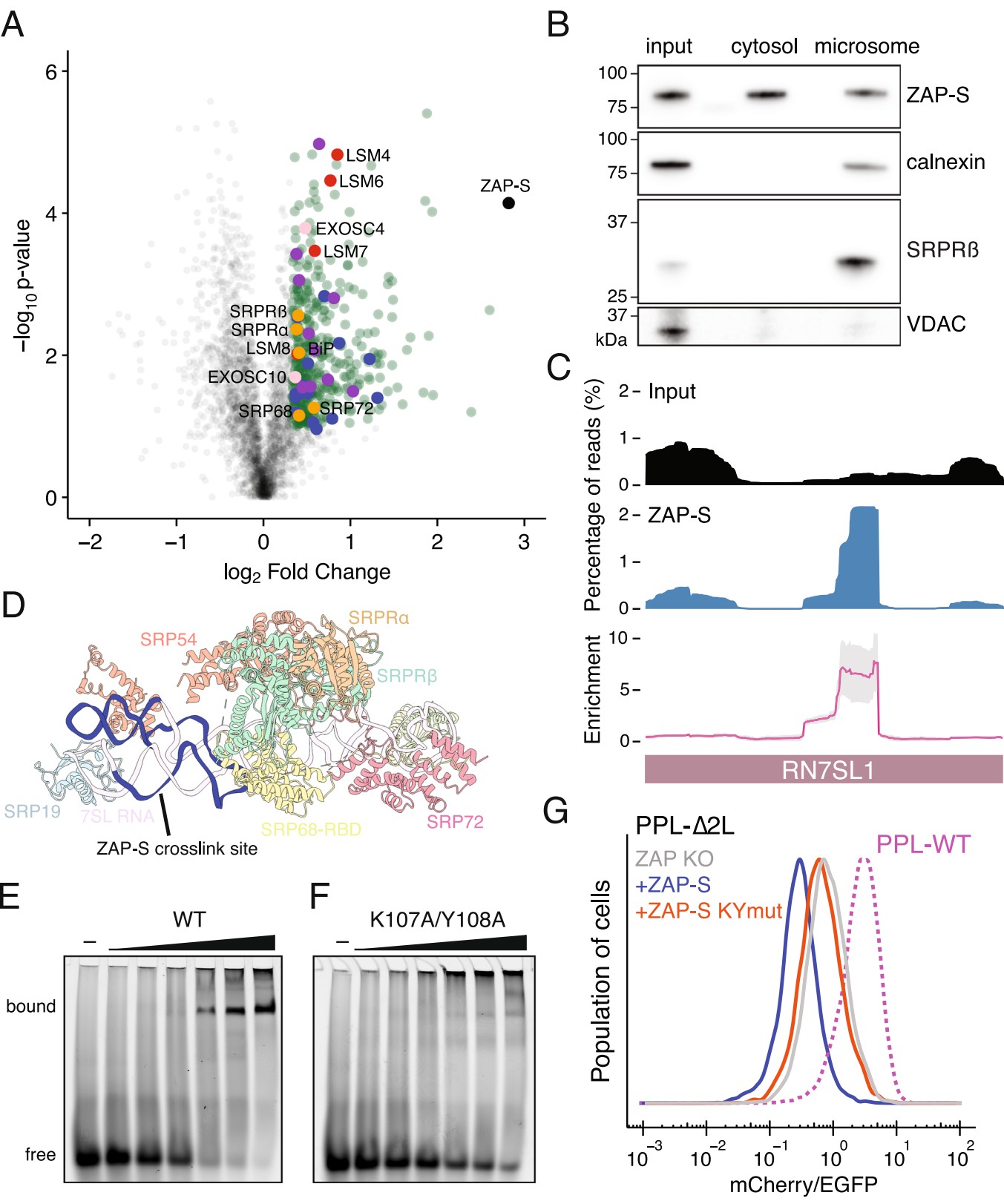

**Figure 4.   ZAP-S promotes RAPP-mediated mRNA decay through its interaction with the SRP.**

(A) Volcano plot analysis of ZAP-S associated proteins. Data represent three biological replicates. Significant proteins are highlighted in green. Dot color corresponds to the enriched Gene Ontology (GO) terms in (Fig. EV4B). Exosome and LSM components are in red and pink, respectively. (B) Immunoblotting analysis of ZAP-S in microsomal fractions isolated from cell expressing the PPL-Δ2 L reporter. Calnexin and VDAC serve as markers for the ER and mitochondria, respectively. Input, whole cell lysates; cytosol, supernatant depleted of microsomes, mitochondria, and nuclei. (C) Gene model of 7SL RNA from input (black) and ZAP-S IP (blue) samples. Bottom: average enrichment of ZAP-S crosslink sites, shown as normalized density (solid line) with standard deviation (shaded area) across two replicates. (D) Structure of the SRP in complex with the SRP receptor (SRPRα and SRPR β) (PDB: 6FRK). ZAP-S crosslink site is colored in blue and indicated. (E, F) Representative EMSA results for ZAP-S WT (E) and K107A/Y108A (F) association with the 7SL RNA probe corresponding to the ZAP-S crosslink site in (D). (G) Flow cytometric analysis of ZAP KO cells expressing the PPL-Δ2 L reporter following rescue with ectopic ZAP-S WT and K107A/Y108A mutant. See also Figure EV4L. Source data are available online for this figure.

## RAPP safeguards ER-targeted mRNAs to prevent UPR activation

ATF5, a key marker for ER stress, is strongly upregulated (~20-fold) in ZAP KO cells (Figs. 5A and EV5A), reaching levels comparable to those observed with thapsigargin treatment (Fig. EV5E). Notably, this upregulation was diminished when expression of the PPL-Δ2 L reporter was shut off by removing doxycycline from the culture media (Fig. EV5F). Addition of doxycycline induced ATF5 upregulation only in ZAP KO but not in parental cells (Fig. EV5F), ruling out a nonspecific effect of doxycycline. While the PPL-Δ2 L reporter remained localized to the ER (Fig. EV5G,H), its abundance within the microsomal fractions was markedly increased in ZAP KO cells (Fig. EV5I). Together, these observations suggest that expression of the PPL-Δ2 L reporter, acting as a RAPP target, triggers the UPR in the absence of ZAP.

To determine whether this UPR activation extends beyond the PPL-Δ2 L reporter, we next tested endogenous candidates predicted to engage the RAPP pathway. We first analyzed differentially expressed genes associated with the GO term—endomembrane and secretory system (Figs. 5A and EV5J), which accounted for ~26% of all upregulated genes in ZAP KO cells. Given that approximately one-third of all human proteins are targeted to the ER, this enrichment likely reflects the intrinsic composition of the human proteome. As a second criterion, we prioritized mRNAs encoding secretory proteins whose signal peptides contain disfavored amino acids (tyrosine, asparagine, glutamine) within their hydrophobic regions. From this group, we selected the signal peptides of CD80 and PRSS8, which contain three and one such residues, respectively (Fig. EV5K).

The selected signal peptides were inserted immediate downstream of the tagBFP initiation codon (Fig. 5E) and the resulting expression constructs were transfected into parental and ZAP-S KO cells. If these mRNAs represent endogenous RAPP substrates, ATF5 upregulation would be expected in ZAP-S KO cells. In line with this prediction, we observed a marked increase in ATF5 mRNA levels in ZAP-S KO cells expressing the signal peptides of CD80 (~50-fold) and PRSS8 (~fivefold), compared to the control lacking a signal peptide (Fig. 5F, ZAP-S KO). In contrast, overexpression of these signal peptides did not induce ATF5 upregulation in parental cells (Fig. 5F, Par). Taken together, these findings support a role for ZAP in the RAPP pathway, regulating the turnover of a subset of ER-targeted mRNAs to maintain ER homeostasis.

## Discussion

Aiming at a comprehensive understanding of the RAPP pathway, here we established a reporter system to study the RAPP quality control pathway and screened for genes that regulate the stability of RAPP mRNA substrates. In addition to regulated mRNA decay, we uncovered the translational repression of RAPP substrate mRNAs (Fig. 6A), a previously uncharacterized feature of the RAPP pathway. Our functional CRISPR genetic screens validated the role of AGO2 in RAPP (Karamyshev et al, 2014) and identified a novel regulator— ZAP. Biochemical and genetic assays revealed that the ZAP-S isoform interacts with the SRP to facilitate RAPP-mediated mRNA decay. When the signal peptide is mutated or SRP54 becomes limiting, SRP engagement becomes non-productive, generating a "faulty" signal. ZAP-S binds the SRP in this non-productive state, an interaction that is essential for RAPP activation. Given that the epistatic relationship between AGO2 and ZAP (Fig. EV3B,C) and previous evidence showing that AGO2 interacts with the mutant signal peptide (Karamyshev et al, 2014), we propose that ZAP-S may function as an adaptor protein to transduce the defective SRP recognition to AGO2-mediated selective mRNA degradation (Fig. 6B). These findings uncover a new function for ZAP-S in the quality control of ER-target mRNAs, protecting cells from the accumulation of potentially mislocalized proteins that could lead to proteotoxic stress.

Our finding that RAPP represses translation of its target mRNAs aligns with observations in other mRNA quality control pathways, such as ribosome collision-induced mRNA degradation and nonsense-mediated decay (NMD). These pathways involve translational repression that can be decoupled from mRNA decay (Hickey et al, 2020; Zinshteyn et al, 2021; Muhlrad and Parker, 1999; Juszkiewicz et al, 2020). While a previous study reported that translational repression is required for ZAP-mediated decay of viral RNAs (Zhu et al, 2012), we found that loss-of-function of ZAP decouples mRNA decay from translational repression during RAPP (Fig. 3D). Thus, we have uncovered a distinct and specific quality control mechanism.

The antiviral activity of ZAP depends on its binding to GNCG motif-containing viral RNAs (Luo et al, 2020). In contrast, our data indicate that ZAP-S association with 7SL RNA is critical for RAPP (Fig. 4C–G). These data suggest that ZAP contributes to the RAPP pathway independently of its antiviral function. Consistent with this notion, ZAP-associated antiviral endoribonucleases, N4BP1 and KHNYN, did not play a role in degrading the RAPP reporter (Fig. EV3F–H). Given that ZAP-S and ZAP-L share an identical N-terminus (Fig. 3E), only ZAP-S, and not ZAP-L, participates in the RAPP pathway. A plausible explanation is that ZAP-L, which is prenylated and membrane-anchored (Charron et al, 2013; Schwerk et al, 2019; Ly et al, 2022), may have limited access to the cytosolic SRP complex, thereby precluding its involvement in RAPP.

While AGO2 is critical for miRNA-mediated translational repression, which is tightly linked to RNA-induced silencing complex

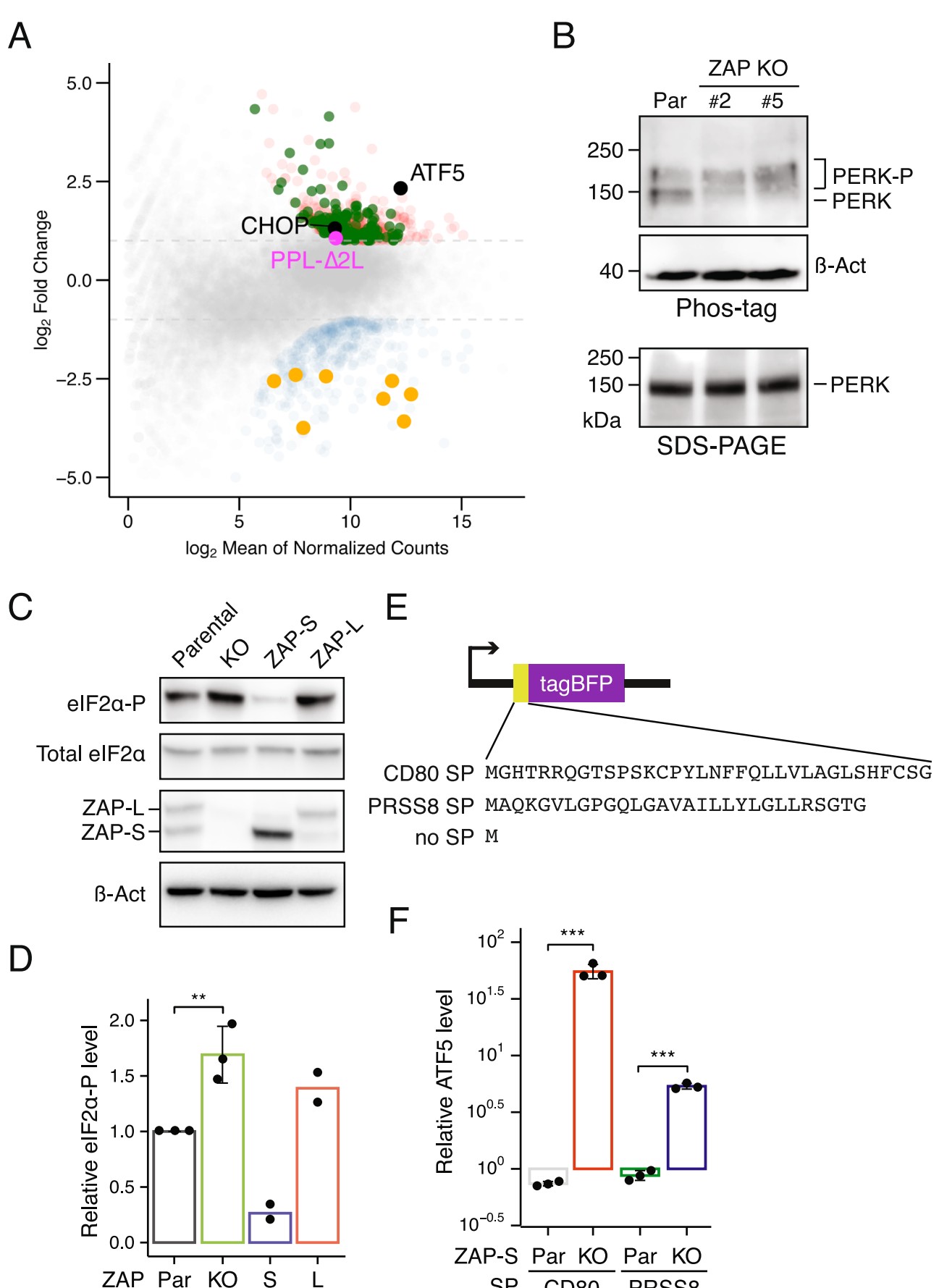

◀

**Figure 5. ZAP-S safeguard ER-targeted proteins to dampen UPR activation.**

(A) MA plot of RNA-seq data comparing ZAP KO and parental cells. Upregulated and downregulated genes are shown in red and blue, respectively. ATF5 and CHOP are highlighted in black, and the PPL-Δ2 L reporter in purple. Orange and green dots denote interferon response-related genes and genes associated with endomembrane and secretory system, respectively. Upregulated genes were subjected to BUSCA subcellular localization prediction (Savojardo et al, 2018). See also Fig. EV5J. (B) Immunoblots for PERK phosphorylation in parental and two ZAKO KO monoclonal lines, resolved by Phos-tag (top) and standard SDS-PAGE (bottom) gels. (C) Representative immunoblots for eIF2α phosphorylation in ZAP KO cells rescued with ectopic expression of ZAP-S or ZAP-L isoform. β-actin serves as a loading control. (D) Quantification of eIF2α phosphorylation shown in (C) (biological replicates, n ≥ 2). Student's t test is indicated by asterisks. **P < 0.01. (E) Schematic representation of signal peptide overexpression constructs. Signal peptides from CD80 and RSPP8 are shown. (F) Relative ATF5 mRNA levels determined by RT-qPCR upon overexpression of the signal peptide constructs shown in (E) in ZAP-S KO and parental cells. Expression levels were normalized to the control expression construct lacking a signal peptide (biological replicates, n = 3). Student's t test is indicated by asterisks. ***P < 0.001. Source data are available online for this figure.

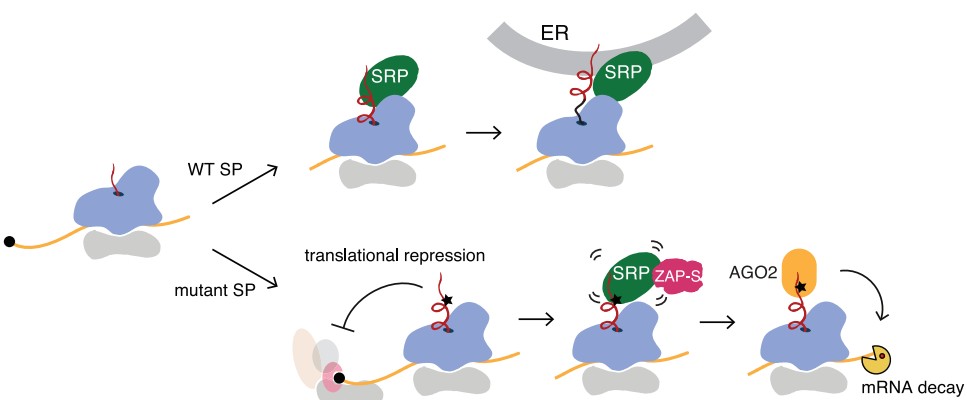

**Figure 6. Proposed model for RAPP-mediated translational repression and mRNA degradation.**

Wild-type signal peptides are efficiently targeted to the ER. Mutant signal peptides engage the SRP non-productively, leading to RAPP-mediated translational repression and mRNA degradation. ZAP-S associates with a non-productive 7SL-containing SRP complex and promotes handoff of the aberrant signal peptide-ribosome complex to AGO2 for mRNA degradation.

(RISC)-mediated mRNA decay (Bartel, 2018), our data indicate that AGO2 is dispensable for RAPP-mediated translational repression (Fig. EV3B,C). This contrast is in line with an earlier observation that the RNA silencing activity of AGO2 is not required for RAPP (Karamyshev et al, 2014), suggesting that its involvement in RAPP is independent of somatic RNA silencing. Interestingly, AGO2 has recently been shown to function in ERAS through its RNA silencing function (Efstathiou et al, 2022). These divergent roles of AGO2 in RAPP and ERAS raise questions about the specific mechanisms governing mRNA decay in each pathway.

While this work aims at understanding RAPP-mediated mRNA decay, it remains to be determined whether the PPL-Δ2 L and PPL-Δ4 L reporters, particularly the SP*-mCherry protein (Fig. 1C), could be eliminated through other protein quality control pathways, such as ER-stress-induced pre-emptive quality control (ER-pQC) (Kang et al, 2006). Given that ER-pQC can target nascent secretory proteins that retain their uncleaved signal peptides (Kadowaki et al, 2015), it is plausible that SP*-mCherry could be a substrate for this pathway. Future studies will aim to identify the factors required for RAPP-mediated translational repression and to further dissect how defective secretory proteins are recognized and degraded at the ER interface.

The UPR is activated in ZAP KO cells even in the absence of the PPL-Δ2 L reporter (~threefold; Fig. EV5F). Moreover, overexpression of the signal peptides from CD80 and PRSS8 was sufficient to further enhance UPR activation (Fig. 5E,F). These results suggest that, in the absence of ZAP, endogenous disfavored secretory proteins that are normally cleared

through RAPP accumulate, resulting in excess protein influx into the ER and thereby overwhelming its protein folding capacity. Furthermore, given that the SRP interacts with N-terminal transmembrane domains as nascent proteins emerge from translating ribosomes (Guna and Hegde, 2018; Hegde and Keenan, 2011) as well as signal peptides, it is conceivable that RAPP-mediated surveillance also regulates mRNAs encoding these domains. Defining the full spectrum of RAPP substrates in future studies will offer further insights into the molecular mechanisms of this quality control pathway. Collectively, our work implies that RAPP broadly impacts the expression of both endogenous and exogenous ER-targeted mRNAs. Further understanding of the mechanisms of RAPP would aid in the optimization of therapeutic protein production, including antibodies and vaccines.

# Methods

**Reagents and tools table**

| Reagent/resource | Reference or source | Identifier or catalog number |
|---|---|---|
| **Experimental models** | | |
| Flp-In T-REx HeLa (*H. sapiens*) | This study | |
| Flp-In T-REx HeLa expressing GRN WT or GRN-A9D reporter | This study | |

| Reagent/resource | Reference or source | Identifier or catalog number |
|---|---|---|
| K562 cells (*H. sapiens*) | ATCC | CCL-243 |
| K562 expressing PPL-WT, PPL-Δ2L or PPL-Δ4L reporter | This study | |
| K562 ZAP KO cells | This study | |
| K562 ZAP-S KO cells | This study | |
| **Recombinant DNA** | | |
| GRN-mCherry-pFRT A9D | This study | |
| GRN-mCherry-pFRT WT | This study | |
| lentiCRISPRv2 | Addgene | 52961 |
| lentiGuide-Hygro | Addgene | 139462 |
| lentiGuide-Puro | Addgene | 52963 |
| p6His-SUMO-HA-Zap-S | This study | |
| pOG44 | Thermo Fisher Scientific | V600520 |
| psPAX2 | Addgene | 12260 |
| pMD2.G | Addgene | 12259 |
| Tet-ON PPL-mCherry PPL-Δ2L | This study | |
| Tet-ON PPL-mCherry PPL-Δ4L | This study | |
| Tet-ON PPL-mCherry WT | This study | |
| **Antibodies** | | |
| Mouse anti-FLAG | Sigma-Aldrich | A8592 |
| Mouse anti-SRP54 | BD Biosciences | 610940 |
| Mouse anti-vinculin | Santa Cruz Biotechnology | sc-73614 |
| Rabbit anti-calnexin | Enzo | ADI-SPA-860 |
| Rabbit anti-calreticulin | Novus Biologicals | NB600-101 |
| Rabbit anti-eIF2α | Cell Signaling Technology | 9722 |
| Rabbit anti-GFP | Clontech | 632381 |
| Rabbit anti-PERK | Cell Signaling Technology | 3192 |
| Rabbit anti-phospho-eIF2α | Abcam | ab32157 |
| Rabbit anti-SRPRB | Bethyl Laboratory | A305-440A |
| Rabbit anti-VDAC | Cell Signaling Technology | 4866 |
| Rabbit anti-ZAP | Proteintech | 16820-1-AP |
| Rabbit anti-β-actin | Cell Signaling Technology | 5125 |
| Rat anti-HA Peroxidase | Sigma-Aldrich | 12013819001 |
| **Software** | | |
| GraphPad Prism 7.0c | GraphPad software Inc | https://www.graphpad.com |
| ImageJ | National Institutes of Health | https://imagej.net/ |
| MAGeCK | Li et al, 2014 | |
| **Other** | | |
| Agilent High Sensitivity DNA Kit | Agilent | 5067-4626 |

| Reagent/resource | Reference or source | Identifier or catalog number |
|---|---|---|
| Anti-HA Magnetic Beads | Thermo Fischer Scientific | 88836 |
| Blasticidin | Invivogen | ant-bl-05 |
| CircLigase ssDNA Ligase | Biosearch Technologies | CL4115K |
| GlycoBlue | Thermo Fischer Scientific | AM9515 |
| high concentration T4 RNA Ligase | New England Biolabs | M0437M |
| Hygromycin B | Thermo Fisher Scientific | 10687010 |
| Microsome Isolation kit | Sigma-Aldrich | MAK340 |
| Murine RNase Inhibitor | New England Biolabs | M0314 |
| NucleoSpin Blood XL and L Kits | Macherey-Nagel | 740950 |
| Phosphatase inhibitor cocktail | Sigma-Aldrich | PHOSS-RO |
| Polybrene | Sigma-Aldrich | TR-1003 |
| PolyJet DNA transfection reagent | SignaGen Laboratories | SL100688 |
| Protease inhibitor cocktail | Sigma-Aldrich | 11836170001 |
| Puromycin | Sigma-Aldrich | P7255 |
| Quick CIP | New England Biolabs | M0525L |
| RNase A | New England Biolabs | T3018 |
| RNase I | Thermo Fischer Scientific | AM2294 |
| SUPERaseIn | Thermo Fischer Scientific | AM2694 |
| Superscript III | Thermo Fischer Scientific | 56575 |
| SuperSignal™ West Femto Maximum Sensitivity Substrate | Thermo Fisher Scientific | 34095 |
| SuperSignal™ West Pico PLUS Chemiluminescent Substrate | Thermo Fisher Scientific | 34580 |
| T4 Polynucleotide Kinase | New England Biolabs | M0201L |
| T4 RNA Ligase 2, Truncated | New England Biolabs | M0242L |
| TRIzol Reagent | Thermo Fischer Scientific | 15596026 |
| TURBO DNase | Thermo Fischer Scientific | AM2238 |
| Zymo-Seq RiboFree total RNA library kit | Zymo Research | R3003 |

## Cell culture conditions

All cell lines were kept and grown at 37 °C in the presence of 5% $CO_2$. K562 cells were grown in RPMI 1640 supplemented with 10% FBS, 2 mM glutamine, and 2 mM sodium pyruvate. HeLa FlpIn

TREx and HEK293T cells were grown in DMEM supplemented with 10% FBS and 2 mM glutamine. Induction of PPL and GRN reporters was carried out by adding doxycycline (Sigma-Aldrich, D9891) at 2 μg/mL for 48 h.

## Generation of reporter cell lines

To create PPL reporter cell lines, K562 cells were transduced with Tet-ON PPL-mCherry WT, Δ2 L, or Δ4 L lentiviral particles and selected with 800 μg/mL G418 (Sigma, G8168) for 2 weeks. Monoclonal lines stably expressing the reporters were isolated by cell sorting and used in all further experiments. To generate GRN reporter cell lines, HeLa FlpIn Trex cells were transduced with GRN-mCherry-pFRT WT or A9D plasmid and pOG44 (Thermo Fisher Scientific, V600520) plasmid according to the manufacturer's instructions.

## Molecular cloning of sgRNAs

Lentiviral plasmids were made by digesting lentiCRISPRv2 (Addgene #52961), lentiGuide-Puro (Addgene #52963), or lentiGuide-Hygro (Addgene #139462) with BsmBI-v2 at 55°C for 30 min at followed by heat inactivation at 80 °C for 20 min. Digested vector was gel-purified then ligated with phosphorylated sgRNA pairs at 1:6 molar ratio at 22 °C for 1.5 h using T4 DNA ligase (New England Biolabs, M0202). Resulting constructs were validated by Sanger sequencing.

## Lentivirus production

One day prior to transfection, 300,000 of HEK293T/17 (ATCC, CRL-11268) cells were plated in a six-well plate and grown in D10 (DMEM supplemented with 2 mM L-glutamine, and 10% fetal bovine serum) media. On the day of transfection, normal growth media was replaced by DMEM supplemented with 2 mM L-glutamine, and 1% fetal bovine serum. In total, 1 μg of lentiviral plasmid was incubated with 600 ng of psPAX2, and 400 ng of pMD2.G and transfected into packaging cells using PolyJet (SignaGen, SL100688). One day post-transfection, cell media was changed to harvest media (DMEM, 1% bovine serum albumin [Sigma, A4612], and 2 mM L-glutamine). Two days post-transfection, lentivirus-containing supernatant was harvested and stored at −80 °C.

## Lentiviral transduction

Lentiviral particles were mixed with K562 cells at 1:1 ratio in a six-well plate and polybrene was added to 8 μg/mL. The cells were centrifuged at 1000 rpm for 2 h at 32 °C. Two days after transduction, cells were ready for hygromycin (400 μg/mL) or puromycin (0.5 μg/mL) selection.

## Generation of cell lines

To generate ZAP KO cells, K562 cells were transduced with lentivirus containing an sgRNA (5'-ATGTGGAGTCTTGAA-CACGG-3') targeting ZAP exon 3. To create ZAP-S isoform-specific KO cells, K562 cells were transduced with lentivirus containing an sgRNA pair targeting ZAP-S iPAS (5'-

TAAAGGAAATGTTGCTGTGG-3' and 5'-TTCATGAA-CATTTTTGGAGT-3'). KO lines were validated by sequencing genomic DNA to characterize Cas9 editing and loss of protein was confirmed by immunoblotting. To rescue ZAP KO cells, coding sequences of ZAP-S, ZAP-S KY mutant, and ZAP-L were cloned into a PiggyBac-Hygro vector. The resulting constructs were transduced into ZAP or ZAP-S KO cells along with a plasmid encoding a PiggyBac transposase using nucleofection (Lonza Bioscience, V4XC-2032). Two days after nucleofection, cells were selected with 400 μg/mL hygromycin B (Thermo Fisher Scientific, 10687010) for 5 days.

## Immunoblot analysis

K562 cells were collected at a density of $1 \times 10^6$ per mL by centrifugation, rinsed once with PBS, and harvested in lysis buffer (20 mM Tris-Cl [pH 8], 150 mM KCl, 15 mM MgCl$_2$, 1 mM DTT, 1% Triton X-100, and 1× cOmplete™ Mini Protease Inhibitor Cocktail [Roche, 11836170001]). HeLa cells were rinsed once with PBS and harvested in the same lysis buffer. Lysates were clarified by centrifugation for 15 min at 14,800 ×g. Equal amount of protein lysates were prepared and boiled in Laemmli loading buffer containing DTT at 95 °C for 5 min. Samples were resolved by 4–12% Criterion XT Bis-Tris protein gels (Bio-Rad, 3450124) or home-made Phos-tag gels as described previously (Wu et al, 2020) and transferred onto PVDF membranes using a Trans-blot Turbo transfer system (Bio-Rad). Membranes were blocked with 5% non-fat milk (Bio-Rad, 1706404XTU) in PBST or TBST for 1 h at RT with gentle rocking. Blots were incubated with indicated primary and secondary (if needed) antibodies according to the manufacturer's instructions. Immunoblots were developed using Super-Signal West Pico Plus ECL substrate (Thermo Fisher Scientific, 34580) supplemented with SuperSignal West Femto Maximum Sensitivity Substrate (Thermo Fisher Scientific, 34095) and imaged with an ImageQuant 800 (Amersham). Band intensities were quantified with ImageJ 1.53 K. Primary antibodies used: FLAG (Sigma-Aldrich, A8592), GFP (Clontech, 632381), ZAP (Protein-tech, 16820-1-AP), β-actin (Cell Signaling Technology, 51255), HA (Sigma-Aldrich, 12013819001), eIF2α (Cell Signaling Technology, 9722), phospho-eIF2α (Abcam, ab32157), SRP54 (BD Biosciences, 610940), vinculin (Santa Cruz Biotechnology, sc-73614), SRPRB (Bethyl Laboratory, A305-440A), VDAC (Cell Signaling Technology, 4866), calnexin (Enzo, ADI-SPA-860), calreticulin (Novus Biologicals, NB600-101), and PERK (Cell Signaling Technology, 3192).

## Flow cytometric CRISPR screen

Lentivirus was generated from the Brunello human CRISPR knockout pooled library (Sanson et al, 2018) (Addgene, #73179) as described before (Coria et al, 2025). To ensure proper Cas9 expression, RAPP reporter cells (PPL-WT and PPL-Δ2 L) were kept under 10 μg/mL of blasticidin (InvivoGen, ant-bl-05) until a day before lentiviral transduction. Two hundred million cells were transduced with the lentivirus-containing media at a MOI of 0.315 with 8 μg/mL of Polybrene (Sigma-Aldrich, TR-1003) as described above. Two days after transduction, cells were selected at 0.5 μg/mL of puromycin (Sigma-Aldrich, P8833) and PPL reporters were induced by adding doxycycline at 2 μg/mL. Six days after

puromycin selection, cells were subjected to cell sorting using a FACSAria III sorter (BD Biosciences). Genomic DNA was extracted from cells using NucleoSpin Blood XL and L Kits (Macherey-Nagel, 740950) according to the manufacturer's protocol. Guide RNA sequences were amplified as described previously (Wu et al, 2020). Purified libraries were quantified with an Agilent Bioanalyzer using a high sensitivity DNA chip (Agilent, 5067-4627) and sequenced on a NextSeq 1000 sequencer (Illumina).

## Analysis of CRISPR screen data

Sequencing reads were aligned to the Brunello library sequences and quantified using custom-written Python scripts. Fold enrichment, false discovery rates, and GO analysis were calculated by MAGeCK (Li et al, 2014) with the following parameters: --gene-lfc-method median --norm-method median --paired.

## Flow cytometry

All flow cytometric data were gathered using a LSRFortessa (BD Biosciences). Shown are representative results from at least two independent experiments.

## Lysate preparation and data analysis for RNA-seq and Ribo-seq

K562 cells expressing the PPL reporters were cultured in T75 flasks and harvested by centrifugation. Cells were washed with PBS once and lysed in 1 mL of footprint lysis buffer (20 mM Tris-Cl [pH 8], 150 mM KCl, 15 mM MgCl$_2$, 1% Triton X-100, 1 mM DTT, 0.1 mg/mL cycloheximide). Lysates were then treated with TURBO DNase (Thermo Fisher Scientific, AM2238) at 2 U/mL for 15 min on ice, then clarified by centrifugation at 21,000×$g$ for 10 min at 4 °C. Clarified lysates were subsequently processed for RNA-Seq or Ribo-seq as described below.

### RNA-seq
Total RNA was extracted using 3 volumes of TRIzol Reagent (Thermo Fisher Scientific, 15596026) according to the manufacturer's instructions, followed by a chloroform extraction. RNA was precipitated from the aqueous phase by with isopropanol, sodium acetate, and GlycoBlue™ (Thermo Fisher Scientific, AM9516), then washed with 70% ice-cold ethanol following precipitation on dry ice. The RNA pellet was resuspended in water and quantified on a Nanodrop 4000. For library preparation, 1.5 μg of total RNA was processed with the Zymo-Seq RiboFree total RNA library kit (Zymo Research, R3003) following manufacturer's protocol. Libraries were quantified and assessed for quality using the Agilent High Sensitivity DNA Kit on 4150 TapeStation system (Agilent 5067-4626) and subsequently loaded onto an Illumina NextSeq 1000 instrument for sequencing.

### Ribo-seq
Ribosome profiling was performed on 20 μg of total RNA from clarified lysates. Samples were digested with 750 units of RNase I (Thermo Fisher Scientific, 2295) for 1 h at 25 °C with gentle agitation. The digestion was stopped by adding 200 units of SUPERaseIn RNase inhibitor (Thermo Fisher Scientific, AM2696). To isolated ribosome-protected footprints, digested lysates were

layered onto a sucrose cushion (20 mM Tris-Cl [pH 8], 150 mM KCl, 5 mM MgCl$_2$, 1 mM DTT, 1 M sucrose) and centrifuged in a TLA100.3 rotor at 100,000 rpm at 4 °C for 1 h. The resulting ribosome pellets were resuspended, and footprints were extracted using miRNeasy mini kit (QIAGEN, 217084). Monosome footprints were size-selected from 15% denaturing TBE-urea gels by excising RNA fragments between 15-40 nt. All subsequent library preparation steps were performed as previously described (Coria et al, 2025). Libraries were sequenced on a NextSeq 1000 machine.

### Data analysis
Ribosome footprints were mapped to an hg19 transcriptome as described before (Coria et al, 2025). RNAseq reads were mapped to the same transcriptome. Ribosome occupancy per transcript for the PPL reporters was calculated by normalizing ribosome footprints to RNA-seq normalized counts (Chothani et al, 2019).

## Immunoprecipitation

In total, 50 million K562 ZAP-S KO cells stably expressing HA-tagged ZAP-S were harvested and lysed in lysis buffer (50 mM HEPES [pH 8], 150 mM NaCl, 5 mM MgCl$_2$, 1% Triton X-100) supplemented with 2U Turbo DNase (Thermo Fisher Scientific, AM2238) as well as protease (Sigma, 11836170001) and phosphatase (Sigma, PHOSS-RO) inhibitor cocktail. In all, 10 μg/mL RNase A (New England Biolabs, T3018) was added to the clarified lysate, followed by 40U murine RNase inhibitor (New England Biolabs, M0314L) to inactive the RNase A. ZAP-S complex was then immunoprecipitated by incubation with pre-equilibrated anti-HA magnetic beads (Thermo Fisher Scientific, 88836) for 1 h at room temperature. The beads were subsequently washed three times with lysis buffer, followed by a final wash with 50 mM HEPES (pH 8), and then resuspended in 30 μL of 50 mM HEPES (pH 8) and processed for mass spectrometry analysis.

## Quantitative mass spectrometry

### Digestion and TMTpro labeling
Sample beads with bound protein were treated with 50 μL of EasyPep lysis buffer and treated with 50 μL of reducing solution and 50 μL of alkylating solution provided with the EasyPep kit (Thermo, A40006) and heated at 95 °C for 10 min at 900 rpm. Samples were allowed to cool to RT and removed the liquid from the beads and treated with 50 μL of 15 ng/μL of Trypsin/LysC and incubated at 37 °C for 4 h. Added 50 μL of 10 μg/μL TMTpro label to each sample (Control → 127 C, 128 C, 129 C; PPL → 131 N, 133 N, 134 N) and incubated at 25 °C overnight. Quenched excess TMTpro with 50 μL of 5% hydroxylamine, 20% formic acid and incubated for 10 min then combined the samples together. Cleaned samples using the EasyPep mini column provided in the kit, eluted in 300 μL of Elution buffer and dried.

### LC/MS analysis
All samples were analyzed on a Dionex U3000 RSLC in front of an Orbitrap Eclipse (Thermo) equipped with an EasySpray ion source with FAIMS™ where indicated. Advanced Peak Determination, Monoisotopic Precursor selection (MIPS), and EASY-IC for internal calibration were enabled and dynamic exclusion was set to a count of 1 for 15 sec in all methods. Solvent A consisted of

0.1% FA in water and Solvent B consisted of 0.1% FA in 80% ACN. Loading pump consisted of Solvent A and was operated at 7 µL/min for the first 6 min of the run then dropped to 2 µL/min when the valve was switched to bring the trap column (Acclaim™ PepMap™ 100 C18 HPLC Column, 3 µm, 75 µm I.D., 2 cm, PN 164535) in-line with the analytical column EasySpray C18 HPLC Column, 2 µm, 75 µm I.D., 25 cm, PN ES902). The gradient pump was operated at a flow rate of 300 nL/min unless otherwise noted.

*LC/MS analysis of IMPDH2 and PREP peptides*

Digested and dried peptides were resuspended in 50 µL of 0.1% FA and 5 µL was analyzed in triplicate using the same linear LC gradient of 5–7% Solvent B for 1 min, 7–30% Solvent B for 84 min, 30–50% Solvent B for 25 min, 50–95% Solvent B for 4 min, holding at 95% Solvent B for 7 min, then re-equilibration of analytical column at 300 nL/min at 5% Solvent B for 17 min. All three injections employed the TopSpeed method with three FAIMS compensation voltages (CVs) and a 1 s cycle time for each CV (3 s cycle time total) that consisted of the following: Spray voltage was 2200 V and ion transfer temperature of 300 °C. MS1 scans were acquired in the Orbitrap with resolution of 120,000, AGC of 4e5 ions, and max injection time of 50 ms, mass range of 350–1600 $m/z$; MS2 scans were acquired in the Orbitrap using TurboTMT method with resolution of 15,000, AGC of 1.25 e5, max injection time of 22 ms, HCD energy of 38%, isolation width of 0.4 Da, intensity threshold of 2.5 e4 and charges 2–6 for MS2 selection. The only difference in the methods was the CVs used, one method used CVs of −45, −60, −75, one used CVs of −50, −65, −80, and the other used CVs of −55, −70, −85.

*Database search*

Raw data files were searched together in Proteome Discoverer 2.4 using the Sequest node. Data was searched against the Uniprot Human database from Feb 2020 using a full tryptic digest, 2 max missed cleavages, minimum peptide length of 6 amino acids, maximum peptide length of 40 amino acids, an MS1 mass tolerance of 10 ppm, MS2 mass tolerance of 0.02 Da, variable oxidation on methionine (+15.995 Da), fixed carbamidomethyl on cysteine (+57.021 Da), and fixed TMTpro modification on lysine and peptide N-terminus (+304.207 Da). TMTpro reporter ions were quantified using the Reporter Ion Quantifier node and normalized on total peptide amount.

## Enhanced UV crosslinking and immunoprecipitation (eCLIP)

eCLIP libraries were generated as described earlier (Blue et al, 2022; Van Nostrand et al, 2016). Briefly, 50 million K562 cells expressing ZAP-S per replicate were washed on ice and collected in cold 1× DPBS and subjected to UV-crosslinking at 400 mJ/cm². Cells were collected by centrifugation, flash frozen in liquid nitrogen, and stored at −80 °C until further processing. Cell lysis was achieved by incubation for 15 min on ice with lysis buffer (50 mM Tris-HCl pH 7.4, 100 mM NaCl, 0.1% SDS) supplemented with protease inhibitor cocktail (Sigma, 11836170001). The cell lysate was treated with RNase I (Thermo Fisher Scientific, AM2295; 1:25 dilution) and Turbo DNase (Thermo Fisher Scientific, AM2238) and incubated at 37 °C for 5 min at 1200 rpm, followed by the addition of murine RNase inhibitor (New England Biolabs, M0314L). The

clarified lysate was transferred to anti-HA magnetic beads (Thermo Fisher Scientific, 88836), subjected to CIP (New England Biolabs, M0525S) treatment at 37 °C for 20 min at 1200 rpm, followed by incubation with PNK at 37 °C for 20 min at 1200 rpm. The protein-bound beads were washed twice with wash buffer (20 mM Tris-HCl [pH 7.4], 10 mM MgCl₂, 0.2% Tween 20), twice with high salt buffer (50 mM Tris-HCl [pH 7.4], 1 M NaCl, 1 mM EDTA, 0.1% SDS) and then finally twice more with wash buffer. A 3' RNA adapter (InvRiL19) was ligated to the samples using high concentration T4 RNA Ligase (New England Biolabs, M0437M) at RT for 75 min. Beads were washed as mentioned earlier and samples eluted by the addition of 1× SDS sample buffer and heating at 80 °C for 10 min. Protein-RNA complex migration was determined by denaturing gel electrophoresis and nitrocellulose membrane transfer. For library preparation, small pieces of the separated complexes were cut out of the membrane and placed in Eppendorf tubes. RNA was released from the membrane by digestion with proteinase K and incubation at 37 °C for 20 min at 1200 rpm, followed by phenol-chloroform extraction and ethanol precipitation. The precipitated RNA was reverse transcribed using Superscript III reverse transcriptase (Thermo Fisher Scientific, 18080093) and InvAR17 primer. Using T4 RNA ligase, a 3' DNA linker (InvRand3Tr3) was ligated to the cDNA product. Libraries were amplified with barcode adapters. PCR products were gel-purified using an 8% native gel and subsequently loaded onto an Illumina NextSeq 1000 instrument for sequencing.

Reads were demultiplexed and trimmed using umi_tools, eclipdemux, and cutadapt described in the Yeo Lab eCLIP pipeline (https://github.com/YeoLab/merge_peaks) (Blue et al, 2022). Reads were aligned to the hg38 reference genome using STAR (Dobin et al, 2013) with relaxed parameters to allow for multimapping reads (outFilterMultimapNmax= 10,000; outFilterMultimapScoreRange= 5). Mapping duplicates were collapsed using umi_tools --method=unique. Peaks were called using the CLIP Tool Kit (Shah et al, 2017) (-p 0.001 --multi-test --valley-seeking --valley-depth 0.90 --minPH 100) and annotated using bedtools intersect against RepeatMasker (rmsk.hg38) and GENCODE v44.

## Overexpression of signal peptide constructs

The signal peptides of CD80 and PRSS8 were cloned into a PiggyBac-Hygro vector and the resulting expression constructs were transduced into ZAP-S KO cells by nucleofection. Cells were harvested two days after nucleofection and subjected to RT-qPCR analysis.

## Isolation of microsomes

The cells were harvested and pelleted by centrifugation at 500 g for 5 min. After a washing step with cold PBS (pH 7.4), microsomes, spherical vesicle-like structures formed from membrane fragments derived from the ER and Golgi apparatus, were isolated using Microsome Isolation kit (Sigma-Aldrich, MAK340). Briefly, cells were resuspended in pre-chilled Homogenization Buffer and then homogenized in a pre-chilled Dounce grinder. Cell lysates were then collected (input) and centrifuged twice at 4 °C to pellet nuclei and mitochondria, first at 10,000× $g$ for 15 min. Supernatant from the first spin was then centrifuged at 10,000× $g$ for 10 min. Supernatant obtained after the second centrifugation, which

contains dilute crude microsomes and cytosolic contents, was centrifuged again at $21,100 \times g$ for 20 min at 4 °C to pellet microsomes. Supernatant (cytosol) was saved and microsomes were gently washed with Homogenization Buffer before being resuspended in SDS buffer for immunoblotting.

## Immunofluorescence microscopy

For K562 cells, chamber slides were pre-coated with poly-L-lysine, and cells were allowed to adhere for 1 h before fixation. Cells were then fixed with 4% paraformaldehyde for 10 min at room temperature and washed three times with PBS. Fixed cells were permeabilized with 0.1% Triton X-100 in PBS for 10 min, followed by three PBS washes. Blocking was performed with 5% BSA in PBS for 1 h at room temperature. Cells were incubated with either anti-HA or anti-FLAG primary antibody (1:200 dilution in 5% BSA) overnight at 4 °C. After incubation, slides were washed three times with PBS containing 0.1% Triton X-100 and incubated with Alexa Fluor–conjugated secondary antibody (1:200 dilution in 5% BSA) for 1 h at room temperature. Cells were washed three times with PBS containing 0.1% Triton X-100 and stained with DAPI (0.1 µg/mL in PBS) for 5 min for nuclear visualization. Slides were mounted and imaged using a Zeiss LSM880 confocal microscope. Z-stack images were acquired and processed using Fiji (ImageJ) software.

## Purification of ZAP-S recombinant protein

The plasmid p6His-SUMO-HA-Zap-S was created by Gibson assembly between the backbone of pSumo-His6-SUMO-AtTPR1(1-209) (Addgene, #177858) and HA-tagged human ZAP-S gene. The protein was expressed in Rosetta (DE3) cells (Novagen) induced with 1 mM IPTG and grown overnight at 16 °C. Cells were lysed by sonication in 50 mM HEPES (pH 7.5), 1 M NaCl and the clarified supernatant was incubated with NiNTA-agarose resin equilibrated in 50 mM HEPES (pH 7.5), 1 M NaCl, 1 mM BME overnight ta 4 °C with gentle rotation. Resin was washed with 10 volumes 50 mM HEPES, pH 7.5, 1 M NaCl, 1 mM BME, 30 mM Imidazole and eluted with 3 volumes of 50 mM HEPES (pH 7.5), 1 M NaCl, 1 mM BME, 500 mM Imidazole. Fractions containing the protein were pooled, concentrated using an Amicon Ultra-15 10KDa MWCO (Sigma-Aldrich, UFC901008D) and applied to a 26/600 Superdex 200 pg column (Cytiva) equilibrated in 50 mM HEPES (pH 7.5), 1 M NaCl. Fractions containing protein without excess nucleic acid were pooled and concentrated as above. The final concentrated protein was dialyzed against 20 mM HEPES, 7.6, 1 M NaCl, 1 mM DTT, 20% glycerol at 4 °C overnight. Small aliquots were frozen in LN2 and stored at −80 °C prior to use.

## Electrophoretic mobility shift assay (EMSA)

Cy5-labeled (5'-Cy5-GGGGACCACCAGGUUGCCUAAG-GAGGGGUGAACCGGCCCAGGUCG-3') and non-labeled RNA that represents the ZAP-S CLIP site was purchased from Millipore Sigma and folded in 10 mM Tris-Cl (pH 7.6), 80 mM KCl by heating to 85 °C and slow cooling to RT. Subsequently, MgCl2 was added to 5 mM, and the RNA was incubated at RT for 20 min. Binding reactions were performed in 20 mM HEPES (pH7.5), 300 mM KCl, 5 mM DTT, 5 mM MgCl2, 20% glycerol and various

concentrations of 6His-SUMO-HA-Zap-S protein at 37 °C for 20 min. Purified protein was diluted in 50 mM Tris (pH 7.5), 25 mM NaCl, 1 mM DTT and 20% glycerol. Ficoll was added to 4% and the sample was loaded onto a 6% polyacrylamide (29:1 Acrylamide:Bis) 1× TBE gel. Imaging was done on an Amersham Typhoon 5 Biomolecular Imager (Cytiva) and quantification of substrate disappearance by ImageQuant TL (Cytiva). Calculation of *Kd* was determined using GraphPad Prism 9 with one-site specific binding model with Hill Slope.

## Statistical analysis

Significance was calculated using unpaired Student's *t* test (*$P < 0.05$, **$P < 0.01$, and ***$P < 0.001$). Error bars denote the standard deviations.

## Data availability

The RNA-seq and Ribo-seq datasets generated in this study have been deposited in the SRA database under accession number: PRJNA1265689. Mass spectrometry datasets in this study have been uploaded to MassIVE under accession number: MSV000099053 (https://massive.ucsd.edu/ProteoSAFe/dataset.jsp?task=485d81850ac842a98e14b7348b6627b0).

The source data of this paper are collected in the following database record: biostudies:S-SCDT-10_1038-S44318-026-00720-4.

## Peer review information

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

## Acknowledgements

We would like to thank Jeff Carrell and Megan Karwan at the NCI-Frederick Flow Cytometry Core Facility for assistance in cell sorting; Ronald Holewinski and Thorkell Andresson at the NCI-Frederick Protein Characterization Laboratory for mass spectrometric analysis; Sabbi Lall (Life Science Editors) for critical reading and editing of the manuscript. This work was supported by the Intramural Research Program of the National Institutes of Health, National Cancer Institute, Center for Cancer Research (1ZIABC012037).

## Author contributions

**Akruti Shah**: Formal analysis; Validation; Investigation; Methodology; Writing—original draft. **Aaztli R Coria**: Formal analysis; Validation; Investigation; Methodology; Writing—original draft. **Britnie Santiago Membréno**: Formal analysis; Validation; Investigation; Methodology; Writing—original draft. **Emilien Orgebin**: Investigation; Methodology. **Jennifer T Miller**: Formal analysis; Investigation; Methodology. **Wilfried Guiblet**: Data curation; Formal analysis; Visualization. **Colin Chih-Chien Wu**: Conceptualization; Formal analysis; Supervision; Funding acquisition; Investigation; Visualization; Writing—original draft; Project administration; Writing—review and editing.

Source data underlying figure panels in this paper may have individual authorship assigned. Where available, figure panel/source data authorship is listed in the following database record: biostudies:S-SCDT-10_1038-S44318-026-00720-4.

## Funding

## Disclosure and competing interests statement

The authors declare no competing interests. This research was supported by the Intramural Research Program of the National Institutes of Health (NIH). The contributions of the NIH author(s) were made as part of their official duties as NIH federal employees, are in compliance with agency policy requirements, and are considered Works of the United States Government. However, the findings and conclusions presented in this paper are those of the authors and do not necessarily reflect the views of the NIH or the U.S. Department of Health and Human Services.

# Expanded View Figures

**Figure EV1.  Establishing a reporter system to study the RAPP quality control pathway.**

(**A**) RT-qPCR analysis of relative mCherry mRNA levels normalized to EGFP mRNA in HeLa cells transduced with the PPL-WT, Δ2 L, or Δ4 L reporter. Error bars indicate standard deviations (biological replicates, $n = 3$). Student's *t* test is indicated by asterisks. $**P < 0.01$. (**B**) Immunoblots for the RAPP reporters in HeLa cells. EGFP serves as an expression and loading control. SP*-mCherry denotes mCherry retaining an uncleaved signal peptide. (**C**) Representative immunofluorescence images of K562 cells expressing the PPL-WT reporter stained with anti-Calreticulin, anti-FALG (PPL-WT reporter), and anti-EGFP antibodies. Colocalization of calreticulin and the PPL-WT reporter is shown. Scale bar, 5 μm. (**D**) Flow cytometric analysis of monoclonal K562 cells expressing the PPL-WT or PPL-Δ2 L reporter. (**E**) Flow cytometric analysis of PPL-Δ2 L reporter cells transduced with sgRNA targeting mCherry compared to a control (Ctrl) sgRNA.

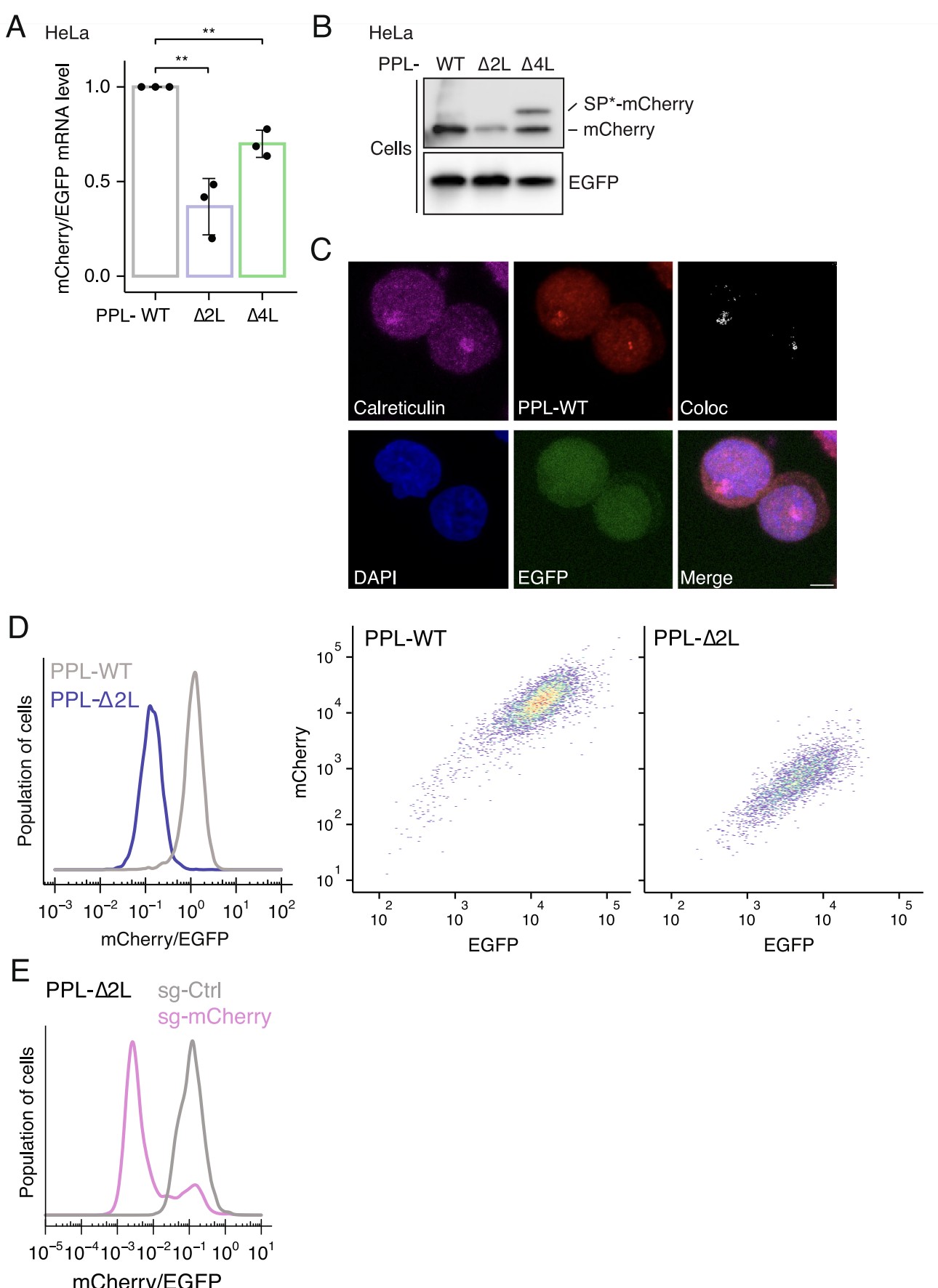

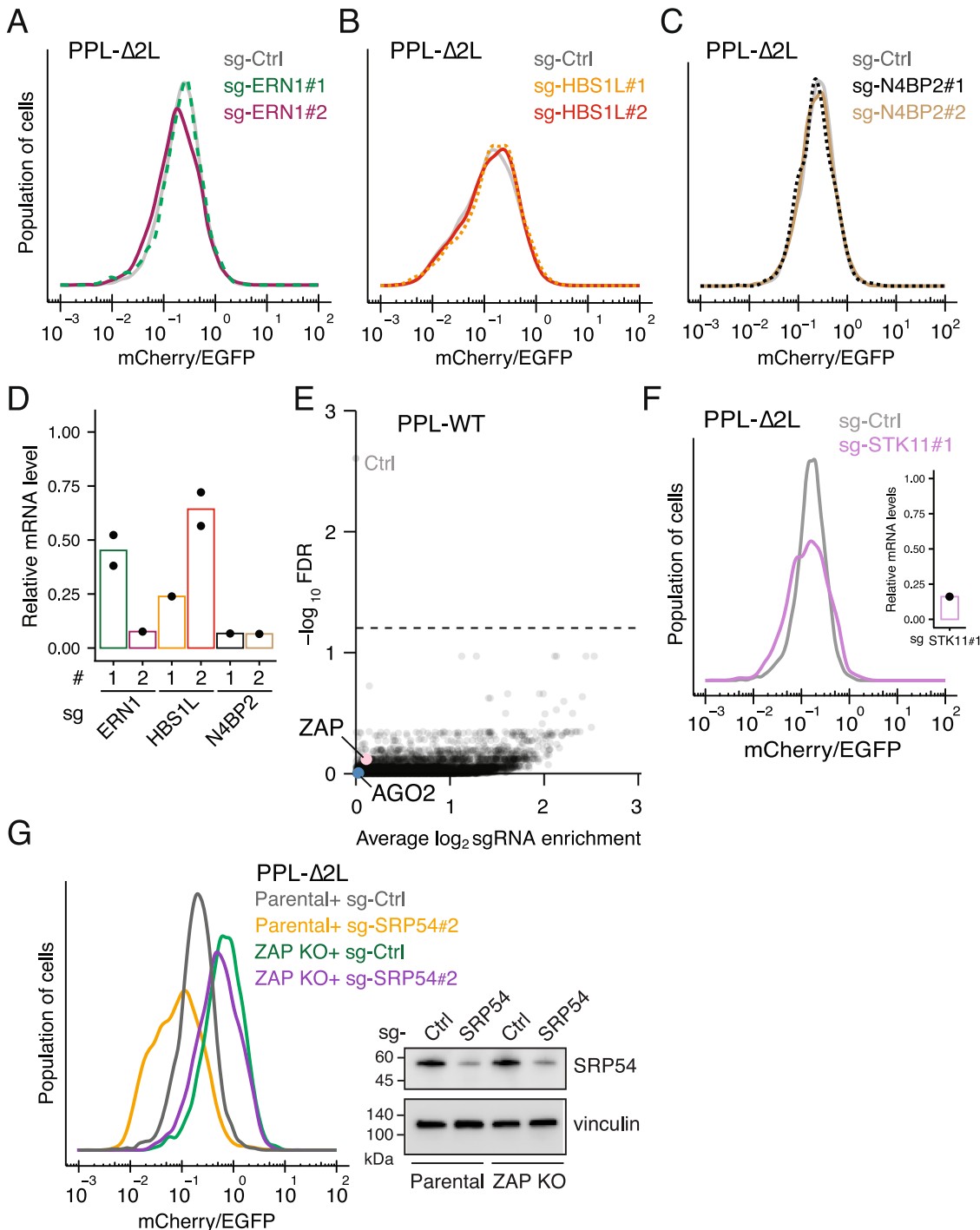

**Figure EV2.  A control genetic screen using the PPL-WT reporter.**

(**A–C**) Flow cytometric analysis of PPL-Δ2 L reporter cells after transduction of sgRNAs targeting ERN1 (**A**), HBS1L (**B**), and N4BP2 (**C**). (**D**) RT-qPCR analysis of target gene expression corresponding to sgRNAs shown in (**A–C**). Each dot represents the mean of three technical replicates (biological replicates, $n \geq 1$). (**E**) A control FACS-based CRISPR screen using the PPL-WT reporter. Volcano plot showing effect size (average $\log_2$ fold change, $x$ axis) versus false discovery rate ($-\log(FDR)$, $y$ axis). AGO2 and ZAP are highlighted in blue and pink, respectively. (**F**) Flow cytometric analysis of PPL-Δ2 L reporter cells after sgRNA-mediated ablation of STK11 compared to a control sgRNA (left). Validation of STK11 knockdown by RT-qPCR (right). (**G**) Flow cytometric analysis of parental and ZAP KO cells expressing the PPL-Δ2 L reporter following depletion of SRP54 (left). Validation of SRP54 knockdown by immunoblotting (right).

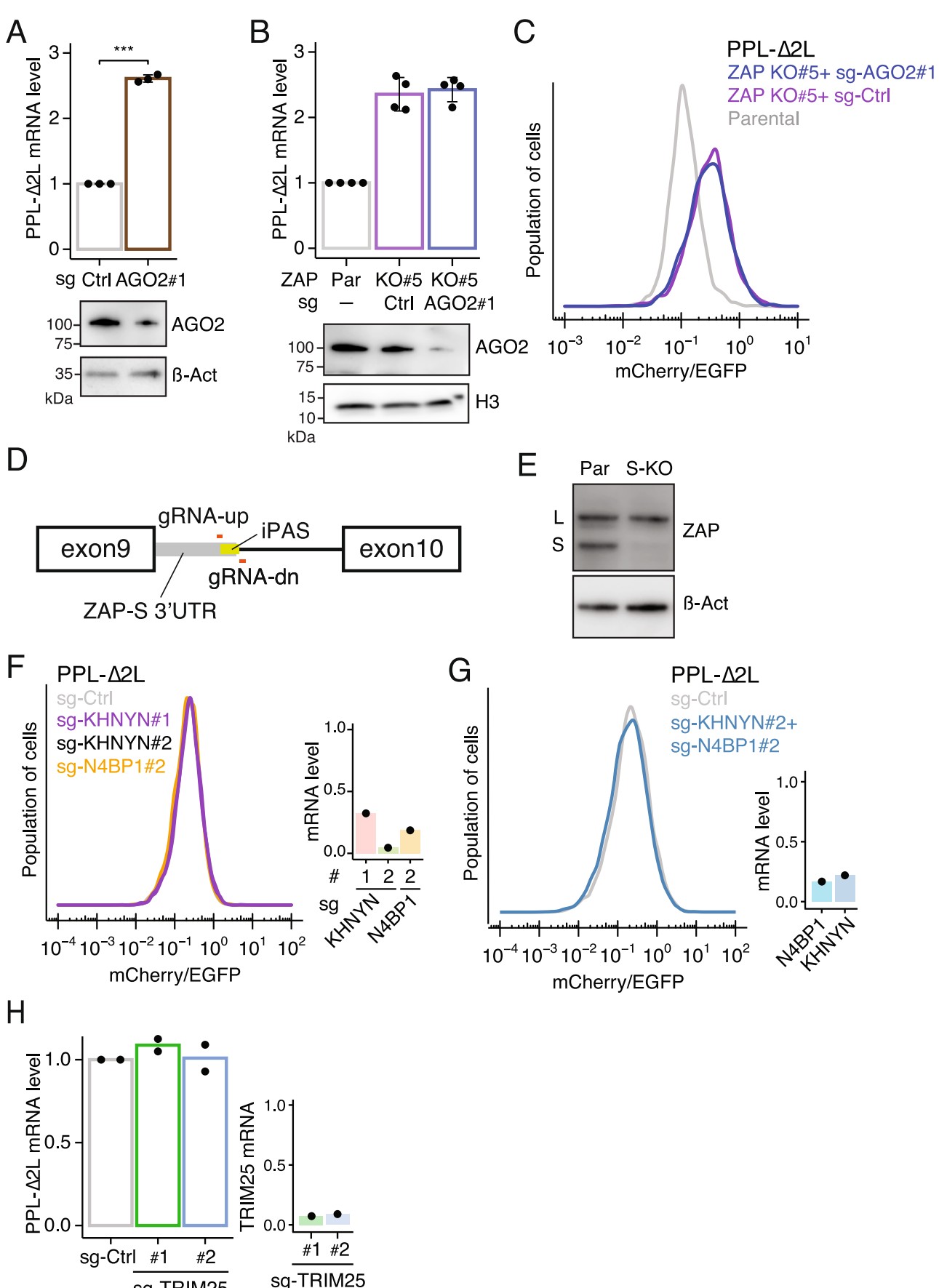

◀  **Figure EV3.  ZAP-S participates in the RAPP pathway.**

(**A**) RT-qPCR assessment of PPL-Δ2 L reporter mRNA following Cas9-mediated AGO2 ablation. Error bars indicate standard deviations (biological replicates, $n = 3$) (top). Student's $t$ test is indicated by asterisks. ***$P < 0.001$. Immunoblots for AGO2 (bottom). (**B**, **C**) RT-qPCR (**B**) and flow cytometric (**C**) analyses of PPL-Δ2 L reporter expression in parental and ZAP KO cells after sgRNA-mediated ablation of AGO2. Error bars indicate standard deviations (biological replicates, $n = 4$). (**D**) Schematic of ZAP exons 9 and 10. The 3′ UTR of ZAP-S isoform is indicated in grey, followed by an intronic polyadenylation signal (iPAS). A gRNA pair (gRNA-up and gRNA-dn) was utilized to remove the iPAS to knock out ZAP-S isoform. (**E**) Immunoblots of parental and a ZAP-S KO isogenic cell line. Two major ZAP isoforms are indicated. (**F**) Flow cytometric analysis of PPL-Δ2 L reporter cells following transduction of sgRNAs targeting N4BP2 or KHNYN compared to a control sgRNA (left). N4BP2 and KHNYN mRNA levels were quantified by RT-qPCR (right). (**G**) Similar to (**F**), PPL-Δ2 L reporter cells were transduced with sgRNAs targeting both N4BP2 and KHNYN (left). Efficiency of N4BP2/KHNYN double depletion was quantified by RT-qPCR (right). (**H**) RT-qPCR analysis of PPL-Δ2 L reporter mRNA levels following Cas9-mediated knockdown of TRIM25 compared to a control sgRNA (left) (biological replicates, $n = 2$). Knockdown efficiency of TRIM25 was assessed by RT-qPCR (right).

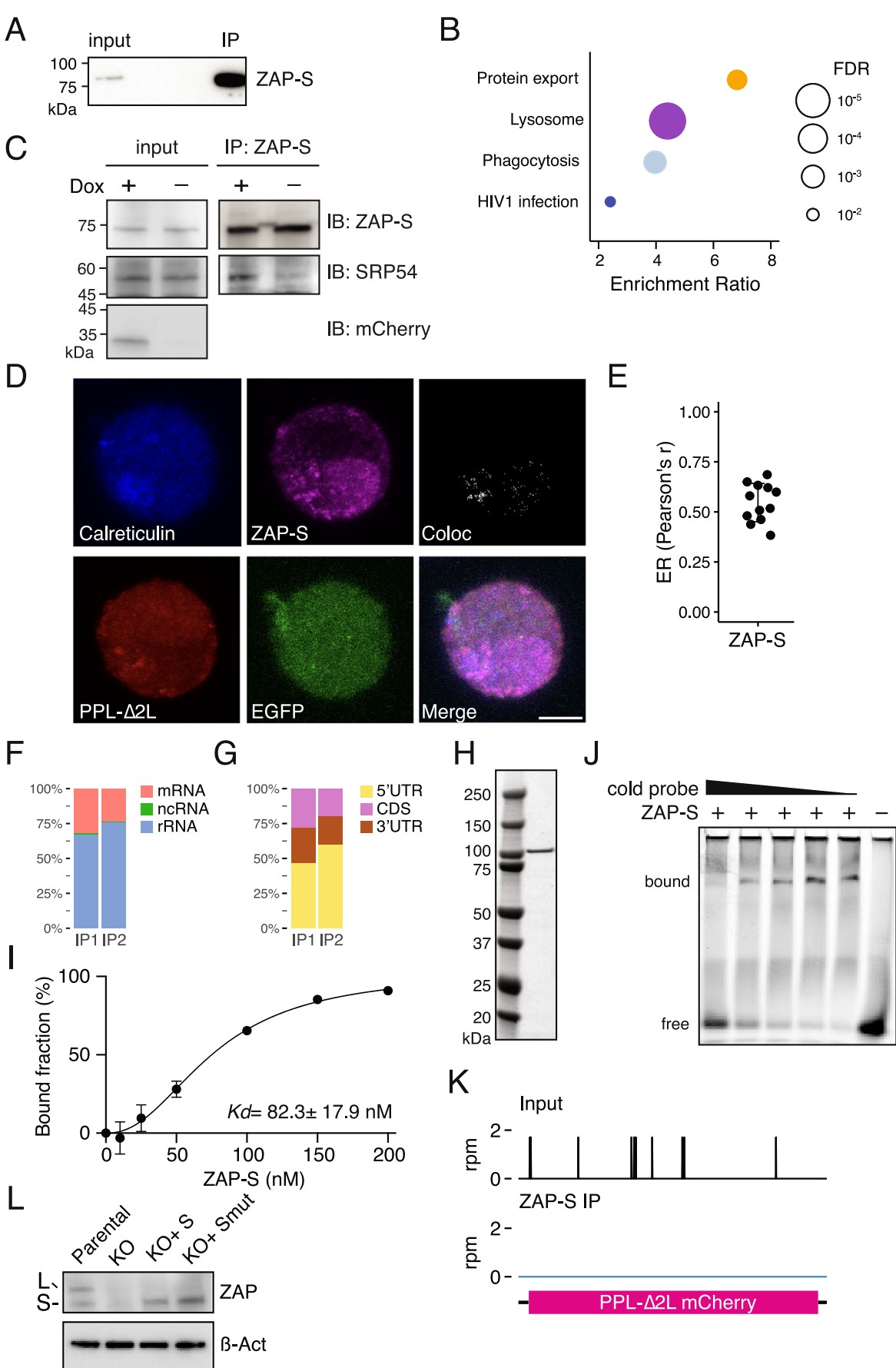

◀ **Figure EV4. ZAP-S facilitates RAPP through its interaction with the SRP.**

(A) Immunoblot for HA-tagged ZAP-S immunoprecipitation. (B) Top four enriched biological processes identified from ZAP-S associated proteins in Fig. 4A. False discovery rate (FDR) is denoted by dot size. (C) Validation of the interaction between ZAP-S and SRP54 by immunoblotting. Expression of the PPL-Δ2 L reporter was induced by Dox (doxycycline). (D) Representative images of K562 cells expressing the PPL-Δ2 L reporter stained with anti-Calreticulin, anti-HA (ZAP-S), anti-FLAG (PPL-Δ2 L reporter), and anti-GFP antibodies. Colocalization of calreticulin and ZAP-S is shown. Scale bar, 5 μm. (E) Quantification of ZAP-S colocalization with the calreticulin-positive pixels using Pearson's r values. Error bars indicate standard deviations ($n = 12$). (F) Composition of reproducible ZAP-S crosslink sites detected in two IP replicates. (G) Distribution of ZAP-S crosslinked reads from (F) across mRNAs (5'UTR, CDS, and 3'UTR). (H) Coomassie staining of purified recombinant ZAP-S protein. (I) EMSA experiment of ZAP-S binding to the 7SL RNA probe. $K$d is indicated ± standard deviations ($n = 3$). (J) Competition EMSA showing specific ZAP-S binding to the 7SL RNA probe. Increasing amounts of unlabeled (cold) 7SL RNA probe were added to compete with labeled 7SL RNA probe. (K) Gene model of the PPL-Δ2 L reporter showing reads from input (black) and ZAP-S IP (blue) samples. (L) Immunoblot for ZAP KO cells rescued with ZAP-S WT or K107A/Y108A mutant.

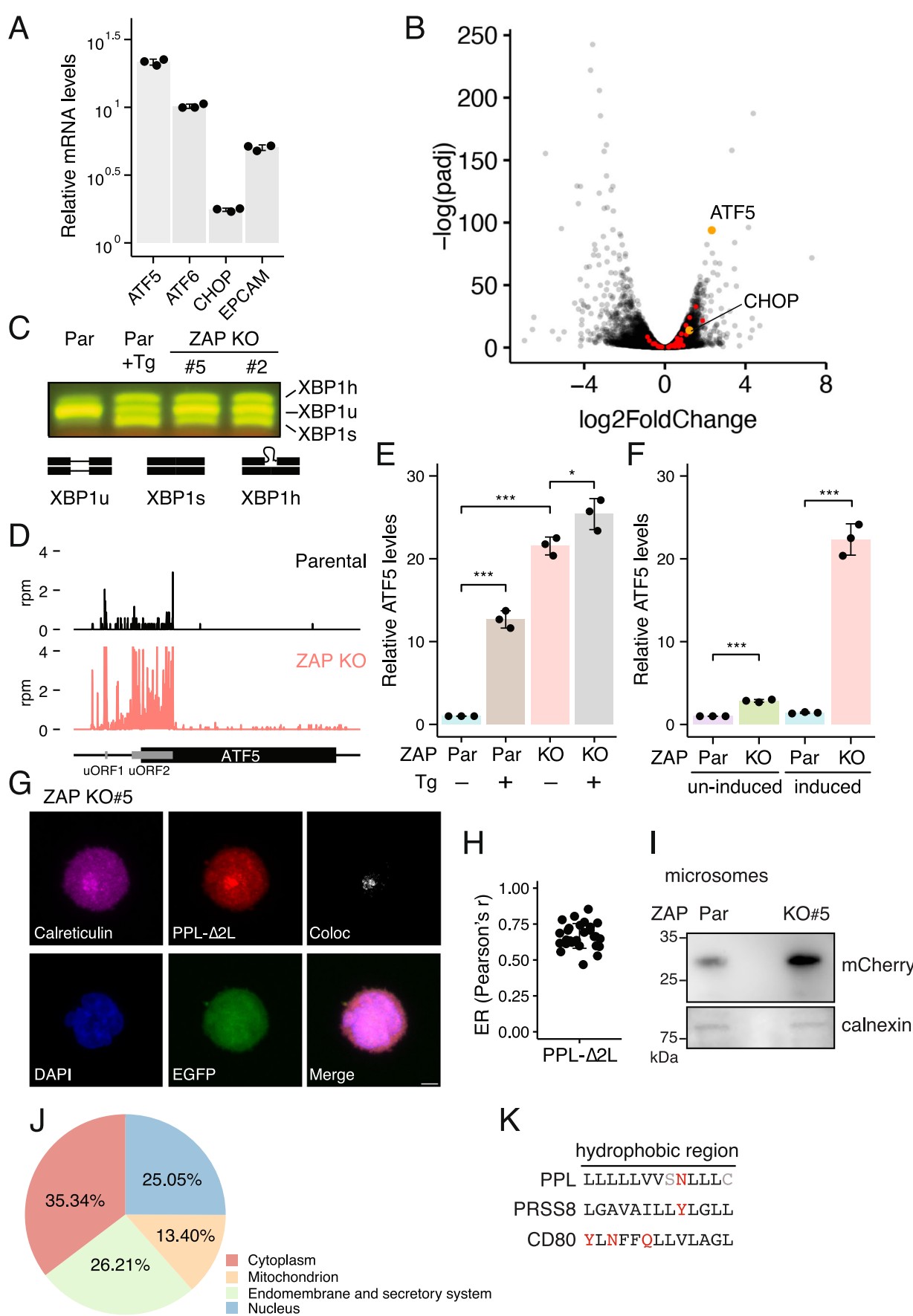

◀ **Figure EV5. Loss-of-function of ZAP induces the UPR.**

(A) Validation of RNA-seq results in ZAP KO cells by RT-qPCR. ATF6, ATF5, CHOP, and EPCAM (a representative secretory protein) mRNA levels are elevated in ZAP KO cells. Error bars indicate standard deviations (biological replicates, $n = 3$). (B) Volcano plot of RNA-seq results comparing ZAP KO to parental cells. ATF5 and CHOP are shown in orange and CHOP-dependent genes in red (biological replicates, $n = 2$). (C) RT-PCR analysis of XBP1 splicing in parental and two isogenic ZAP KO lines. Thapsigargin (Tg) treatment (2 µM for 24 h) was used as a positive control for UPR activation. (D) Ribosome footprint density across ATF5 from parental and ZAP KO cells. uORFs are annotated in grey. (E) RT-qPCR analysis of ATF5 expression levels in parental and ZAP KO cells after thapsigargin (Tg) treatment (2 µM for 24 h). Cells were cultured in the presence of doxycycline to induce the PPL-Δ2 L reporter. Error bars indicate standard deviations (biological replicates, $n = 3$). Student's *t* test is indicated by asterisks. *$P < 0.05$; ***$P < 0.001$. (F) RT-qPCR analysis of ATF5 expression levels in parental and ZAP KO cells with or without induction of the PPL-Δ2 L reporter by doxycycline. Error bars indicate standard deviations (biological replicates, $n = 3$). Student's *t* test is indicated by asterisks. ***$P < 0.001$. (G) Representative images of K562 ZAP KO cells expressing the PPL-Δ2 L reporter stained with anti-Calreticulin, anti-FLAG (PPL-Δ2 L reporter), and anti-GFP antibodies. Colocalization of calreticulin and the PPL-2L reporter is shown. Scale bar, 5 µm. (H) Quantification of PPL-Δ2 L colocalization with the calreticulin-positive pixels using Pearson's r values ($n = 27$). (I) Immunoblots for the PPL-Δ2 L reporter from parental and ZAP KO microsomal fractions. Calnexin serves as an ER marker. (J) BUSCA subcellular localization analysis of upregulated genes in ZAP KO cells. Endomembrane and secretory system includes extracellular space (GO:0005615), endomembrane system (GO:0012505), plasma membrane (GO:0005886), and organelle membrane (GO:0031090). (K) Hydrophobic regions of PPL, PRSS8, and CD80 signal peptides. Hydrophobic residues are shown in black, disfavored amino acids in red, and all other residues in grey.

