## [Peer Review File · The EMBO Journal]

ZAP targets aberrant mRNA transcripts encoding proteins with defective signal peptides for degradation

Akruti Shah, Aaztli Coria, Britnie Membreno, Emilien Orgebin, Jennifer Miller, Wilfried Guiblet, and Colin Wu

Corresponding author(s): Colin Wu (colin.wu2@nih.gov)

Review Timeline:

Submission Date:	8th Jun 25
Editorial Decision:	9th Jul 25
Revision Received:	24th Oct 25
Editorial Decision:	10th Dec 25
Revision Received:	23rd Dec 25
Accepted:	16th Jan 26

Editor: Ioannis Papaioannou

Transaction Report:

Dear Dr. Wu,

Thank you for submitting your manuscript EMBOJ-2025-121583 for consideration by The EMBO Journal, and for your patience during peer review. Your manuscript has now been seen by four experts in the field, and we have received their informative and constructive reports, which you can find below.

I am pleased to say that, as you will see, the referees indicate interest in your manuscript, find your results intriguing and likely interesting to the field, the study for the most part well-performed, and the manuscript well-written. However, they also point out that further validation of some reagents/observations, stronger evidence to support certain conclusions, and further clarification of the model would be necessary before we could publish the manuscript in The EMBO Journal. They list a number of concerns and provide several suggestions for strengthening the work and the manuscript further, which we find largely reasonable and constructive, and likely to increase the impact of the work on the field.

Given the referees' supportive comments and positive recommendations, I would like to invite you to submit a thoroughly revised version of your manuscript taking the referees' suggestions on board, along with a detailed point-by-point response addressing all referees' comments. I should add that it is The EMBO Journal policy to allow only a single round of major revision, and acceptance of your manuscript will therefore depend on the completeness of your responses in this revised version. Please let me know if you have any questions or comments that you would like to discuss with me. If there are any major points you do not agree with or cannot address during your revision, I would encourage you to share them with me as early as possible to discuss how to proceed further in the most efficient way. I should also add that all technical concerns must be adequately addressed.

We generally allow three months as standard revision time (October 8, 2025). As a matter of policy, competing manuscripts published during this period will not negatively impact our assessment of the conceptual advance presented by your study. However, we request that you contact us as soon as possible upon publication of any related work, to discuss how to proceed. Should you foresee a problem in meeting this three-month deadline, please let us know in advance and we will be able to grant an extension.

Thank you for the opportunity to consider your work for publication in The EMBO Journal. I look forward to your revision.

Best regards,

Ioannis

Instructions for preparing your revised manuscript

1. When you are ready to submit the revision, please upload:

- A Word file of the manuscript text (including legends of main Figures, EV Figures and Tables). Please make sure that changes are highlighted (or "tracked") to be clearly visible.

- Individual production-quality figure files (one file per figure). When assembling your figures, please refer to our figure preparation guidelines in order to ensure proper formatting and readability in print as well as on screen:

If the data shown in a figure are obtained from n {less than or equal to} 2, please use scatter plots showing the individual data points.

- i. the name of the statistical test used to generate error bars and P values
- ii. the number (n) of independent experiments (please specify technical or biological replicates) underlying each data point (discussion of statistical methodology can be reported in the Materials and Methods section, but figure legends should contain a basic description of n , P , and the test applied)
- iii. the nature of the bars and error bars (s.d., s.e.m.).

- A point-by-point response to the referees' comments, with a detailed description of the changes made (as a word file). All referees' concerns must be fully addressed and their suggestions taken on board. When preparing your letter of response to the referees' comments, please bear in mind that this will form part of the Review Process File and will therefore be available online to the community. Please note that you have the possibility to opt out of the transparent process at any stage prior to publication by letting the editorial office know (contact@embojournal.org); if you do opt out, the Review Process File link will point to the following statement: "No Review Process File is available with this article, as the authors have chosen not to make the review process public in this case.". For more details on our Transparent Editorial Process, please visit our website: <https://www.embopress.org/page/journal/14602075/authorguide#transparentprocess>

- Expanded View (EV) files (replacing Supplementary Information) that are collapsible/expandable online. A maximum of 5 EV Figures can be typeset. EV Figures should be cited as "Figure EV1, Figure EV2" etc. in the text, and their respective legends should be included in the manuscript file after the legends of regular figures. See detailed instructions regarding Expanded View files here: <https://www.embopress.org/page/journal/14602075/authorguide#expandedview>

- For the figures that you do NOT wish to display as Expanded View figures, they should be bundled together with their legends in a single PDF file called "Appendix", which should start with a short Table of Contents (including page numbers). Appendix figures should be referred to in the main text as: "Appendix Figure S1, Appendix Figure S2" etc. Please see detailed instructions here: <https://www.embopress.org/page/journal/14602075/authorguide#expandedview>

- A complete author checklist, which you can download from our author guidelines (<https://www.embopress.org/page/journal/14602075/authorguide>). Please note that the checklist will also be part of the Review Process File.

2. Please note that no statistics should be calculated and shown in Figures if $n=2$. Please also note that each p value should be reported as an exact value.

3. Before submitting your revision, primary datasets (and computer code, where appropriate) produced in this study need to be deposited in appropriate public databases (see <https://www.embopress.org/page/journal/14602075/authorguide#dataavailability>). In particular, you are kindly requested to deposit all DNA and RNA sequencing data, as well as the mass spectrometry data generated in the study. The accession numbers, database, and the specific URLs (links) should be listed in a formal "Data availability" section (placed after Methods), following the example below:

"The RNA-seq datasets produced in this study are available in the following database:
Gene Expression Omnibus GSE46843 (<https://www.ncbi.nlm.nih.gov/geo/query/acc.cgi?acc=GSE46843>)"

*** All links should resolve to a page where the data can be accessed. ***

*** Please remember to provide in the Data availability section of your revised manuscript reviewer passwords if the datasets are not yet public. ***

*** The Data Availability Section is restricted to new primary data that are part of this study. In case you have no data that require deposition in a public database, please state so instead of referring to the database: "Our study includes no data deposited in public repositories." under the heading "Data availability". ***

4. The materials and methods need to be described in the manuscript using our structured methods format, which is now required for all research articles. According to this format, the Methods section includes a single "Reagents and Tools Table" - listing key reagents, experimental models, software and relevant equipment including their sources and relevant identifiers - followed by a "Methods and Protocols" section describing the methods. Please download and fill our Reagents and Tools Table template (.docx), which you can find in our author guide: <https://www.embopress.org/page/journal/14602075/authorguide#structuredmethods>. When submitting your revised manuscript, please do not include the Reagents and Tools Table in the Methods section of the manuscript but instead upload it as a separate file choosing the file type "Reagent Table".

5. Please check that the title and the abstract of the manuscript are brief, yet explicit, even to non-specialists. The length of the title should not exceed 100 characters, and the abstract should be a single paragraph not exceeding 175 words.

6. Please also note our reference format: <https://www.embopress.org/page/journal/14602075/authorguide#referencesformat>.

8. Please remember: digital image enhancement is acceptable practice, as long as it accurately represents the original data and conforms to community standards. If a figure has been subjected to significant electronic manipulation, this must be noted in the

figure legend or in the "Materials and Methods" section. The editors reserve the right to request original versions of figures and the original images that were used to assemble the figure.

9. Our journal encourages inclusion of data citations in the reference list to directly cite datasets that were obtained from public databases. Data citations in the article text are distinct from normal bibliographical citations and should directly link to the database records from which the data can be accessed. In the main text, data citations are formatted as follows: "Data ref: Smith et al, 2001" or "Data ref: NCBI Sequence Read Archive PRJNA342805, 2017". In the Reference list, data citations must be labeled with "[DATASET]". A data reference must provide the database name, accession number/identifiers, and a resolvable link to the landing page from which the data can be accessed at the end of the reference. Further instructions are available at: <https://www.embopress.org/page/journal/14602075/authorguide#referencesformat>.

10. We request authors to consider both actual and perceived competing interests. Please review our policy (<https://www.embopress.org/page/journal/14602075/authorguide#conflictsofinterest>) and update your competing interests statement if necessary. Please name this section 'Disclosure and competing interests statement' and place it after the Acknowledgements section.

11. Please note that all corresponding authors are required to provide an ORCID ID upon submission of a revised manuscript (<https://orcid.org/>). Please find instructions on how to link your ORCID ID to your account in our manuscript tracking system in our Author guidelines (<https://www.embopress.org/page/journal/14602075/authorguide#authorshipguidelines>).

12. We use CRediT to specify the contributions of each author in the journal submission system. CRediT replaces the author contribution section, which should be removed from the manuscript. Please use the free text box to provide more detailed descriptions. See also guide to authors: <https://www.embopress.org/page/journal/14602075/authorguide#authorshipguidelines>.

14. We would also welcome the submission of cover suggestions or motifs to be used by our Graphics Illustrator in designing a cover.

15. Please use the link below to submit your revision:

Referee #1:

Summary

Regulation of Aberrant Protein Production (RAPP) is a quality control pathway that degrades the nascent chains with aberrant signal peptides, as well as the translating mRNA. While this pathway was first described in 2014, there are still major gaps in our understanding of how these aberrant mRNA/nascent chains are recognized and degraded. In this study, the authors used a functional genetic screen to identify zinc finger antiviral protein (ZAP) as a key component of this pathway. Overall, this study is well-designed, informative, and will be of interest for EMBO J readers. However, there are some concerns about validation of key reagents, as well as some confusion about how ZAP fits into the existing model of RAPP.

Major comments:

1. Overall- sgRNAs that are used, outside of those in the screen, should be validated either by WB or qPCR. In particular the ZAP isoform specific sgRNAs should be validated by WB.
2. The authors should make it clear in the results section the cell type they are using (K562) and give a brief rationale for the choice.
3. It is a little disconcerting that the PPLdel4L is only minimally reduced, unlike what was reported in the original publications. The rationale proposed (complete loss of SRP interaction) is a reasonable one but the citations given (line 116) do not support this rationale, and it isn't clear why this would vary between different cell lines. Also, this hypothesis that the higher MW PPL band represents cytosolic PPL should be confirmed by ICC or fractionation.
4. For all FACS figures- it would be very helpful to plot the WT and PPLdel2L together so that they can be directly compared.
5. The use of translation efficiency in this context
6. In 3A the blot should also be probed for PPL to see whether there is any protein production. ICC to determine whether it is in the secretory pathway or cytosolic would also be interesting.
7. In 3D- it is confusing that the TE of PPLdel2L isn't increased with ZAP knockout. My understanding of the model is that the

reduced TE of PPLdel2L is because translation triggers degradation; therefore transcripts with ribosomes are depleted compared to those without ribosomes. If this degradation is impeded, then that ratio should be rescued. Can the authors comment on this.

8. Method of biochemical fractionation needs to be included in methods section

9. EV4B immunoblot needs some additional controls to demonstrate the purity of this fraction- e.g nuclear and mitochondrial markers. Would also be helpful to use a more conventional ER marker, that isn't a known interactor of ZAP-S, eg calretinin. You should also confirm this finding on imaging- ideally staining for endogenous ZAP-S, but if those antibodies are not suitable for ICC you should at least stain for your tagged exogenous ZAP-S and confirm it is in the ER by imaging.

10. WB in fig 5B needs loading control

11. Overall, I find it challenging to conceptualize how ZAP fits in the existing model of RAPP. The original model postulates that signal sequences which fail to interact with SRP come into close proximity of Ago2 and are then degraded. In this model, signal sequences must have at least a weak interaction with SRP in order to be degraded by RAPP (since ZAP is actually targeted to the mutant RNA/nascent chain through SRP). However, both in the original study, as well as in this study, knockdown of SRP promotes robust degradation. Is this process (degradation after SRP54 knockdown) ZAP dependent? I think this is a critical experiment.

Minor comments

1. Chothani et al 2019 citation (in methods) is not in references

Referee #2:

Shah et al analyzed how RAPP is regulated by ZAP-S. They first set up a reporter for RAPP in which a signal peptide was added to mCherry. A version of this with two leucines deleted led to decreased mCherry expression. The authors then used a CRISPR screen to identify proteins required for RAPP. ZAP was one of the three hits in this screen and its depletion increased reporter expression. It appears that ZAP is required for RAPP reporter RNA decay and not translational repression. The authors then show that ZAP-S depletion, but not ZAP-L, increases reporter expression. To determine how ZAP-S regulates RAPP, they use a combination of CLIP and proteomics to identify that ZAP interacts with the SRP and binds 7SL RNA. They then propose that when ZAP is not present, UPR is triggered because RAPP-mediated quality control is not functional. While the authors show that ZAP regulates RAPP in the context of signal peptides fused to reporter proteins, the mechanism for how ZAP promotes degradation of this RNA is not clear. Overall, this is an exciting paper that shows ZAP is required for RAPP but some of the conclusions need additional data to become more convincing.

Major comments:

1. Figure EV3E tests whether N4BP1 or KHNYN depletion increase reporter expression similar to ZAP. While knocking down either alone has no effect, the authors did not test co-depleting them. This needs to be done to determine if they have functionally redundant activity, which has previously been shown for HIV and LNP-mRNA (Kim et al 2025 and doi: 10.1038/s41467-024-55192-z). The authors should also test whether TRIM25 depletion increases reporter mRNA abundance to determine how ZAP's role in RAPP compares to its antiviral functions. The authors make a point that ZAP contributes to RAPP independent of its antiviral activity, but without testing the role of TRIM25 and a double knockdown for N4BP1 and KHNYN, this has not been robustly tested.

2. The authors suggest that Figure EV4B shows that ZAP-S localizes to the ER. This biochemical fractionation is not very convincing. Immunofluorescence experiments should be used to test whether ZAP-S colocalizes with the ER in these experimental conditions. It should be noted that this has not been observed in several previous studies. ZAP-L localizes to cytoplasmic membranes, including the ER, but ZAP-S is usually diffuse in the cytoplasm. Similarly, the SRP proteins highlighted in Figure 4A represent a small number of the hundreds of proteins identified in the proteomics experiments (Table EV3). This is not strong evidence that ZAP binds the SRP.

3. The CLIP experiments identified a ZAP binding peak on RN7SL1. To validate this, the authors performed an EMSA (Figure 4E-F). While this showed that ZAP can bind the RNA tested, there was not a negative control RNA. Therefore, this experiment does not show that ZAP specifically binds the peak identified by CLIP. Competition EMSAs, or a similar technique, should be used to show that ZAP binds this region with specificity. Overall, the specificity of ZAP for the SRP is not clear. Immunoprecipitating ZAP and testing whether its interaction with SRP components and the 7SL RNA by western blotting and RT-qPCR, respectively, would enhance confidence in these findings.

Minor comments:

1. Figure 1 analyzed how the mutated signal peptide inhibits mCherry expression. The authors say that Figures 1D and 1E show that translation is inhibited but this is not clear from the data. There is at best only a very small effect in the polysome profiling experiment shown in Figure 1D. The data in Figures 1E and EV1A are normalized to the point where it is not clear what the real effect is on mCherry expression. The RNA analysis in EV1A needs to be shown as normalized counts with mCherry and EGFP shown independently. The Ribo-seq data needs to be shown clearly so the ribosome coverage across the mCherry and GFP

transcripts can be compared.

2. Similar to minor comment 1, the RNA-seq data in Figure EV3A and the Ribo-seq data in Figure 3D need to be shown as normalized counts and coverage plots to visualize the data appropriately.
3. The authors report that ZAP-S depletion increased mCherry expression, but ZAP-L depletion had little effect (Figure 3F). Figure EV3C shows specific ZAP-S knockout but I didn't see a figure showing the ZAP-L was specifically depleted. The authors need to show this.
4. The RT-qPCR results in Figure 3J shows the wild-type GRN signal peptide controls. Figure 3B, 3H and EV3A do not show the wild-type PPL signal peptide control. In figure 3J, it is clear that ZAP-S knockdown leads to a partial rescue of reporter RNA abundance. For Figure 3B, 3H and EV3A it is unclear if ZAP depletion leads to a full or partial rescue and the wild type control needs to be present to determine this.

Referee #3:

RAPP, a mechanism of mRNA degradation/translation repression for ER-translocation defective proteins, has been proposed by earlier studies. However, the mechanistic details and triggering factors remained elusive. The authors nicely showed that ZAP-S facilitates the pathway associated with 7SL RNA in the SRP complex. This reviewer recommends that the authors address the following points before publication.

Major points

1. The authors nicely showed that ZAP-S binds to 7SL RNA by eCLIP-Seq. This reviewer wonders whether this is constitutive or inducible with the RAPP substrate presence. If the interaction is constitutive, the authors should explain why the RAPP is not always induced for all the mRNA translated on the ER.
2. The authors should test ZAP-S localization on the ER by microscopic methods.
3. In lines 316-317, the logic of " These findings suggest that many ER-targeted genes that were upregulated in ZAP KO cells may represent endogenous cellular substrates of the RAPP pathway" does not make sense. It might be possible, however, data does not support it.
4. Related to point 4, the experiment of Figure 5D, using the natural context of signal sequence, evokes another question: how does ZAP-S discriminate the target? Given the overexpression of the reporter, this could be considered as the overload of an unfavored substrate may lead to RAPP (in that case, another question is why this does not occur in PPL WT reporter). Although this reviewer agreed that Figure 5D is interesting, it requires a solid explanation/experiments/analysis of why these substrates are selected for RAPP.

Minor points

1. While the authors show that ZAP knockout evokes the integrated stress response as evidenced by RNA-Seq and Western blotting, it remains unclear whether the increased translational efficiency of stress-responsive genes, such as ATF4, is also reflected in the ribosome profiling data. Similarly, XBP1 splicing pattern could be retrieved by RNA-Seq and ribosome profiling data.
2. The authors argue that ZAP-S may play a broader role in the RAPP pathway than AGO2, based on their observation that ZAP-S, but not AGO2, regulates the stability of GRN mRNA. However, this conclusion is drawn solely from data on GRN, which represents a limited basis for such a general claim. Therefore, the strength of this assertion should be toned down.
3. In the CRISPR screening section, the main text states that the top 15% of cells with an increased mCherry/EGFP ratio were sorted, whereas the figure legend indicates that the top 5% were sorted. This discrepancy should be addressed, and either the text or the figure should be corrected for consistency.
4. Figure 1D may require a statistical test for the authors' claim.
5. At line 200, the authors may cite the latest reviews for RNA silencing.
6. To further support Figure EV3B, qPCR of reporter mRNAs should be conducted upon AGO2 knockdown in both parental and ZAP KO cells.
7. ZAP-L KO cells were not well-characterized in this paper. The Western blot and others should be shown.

8. In Figure 4A-B and lines 245-255, the authors described the interaction of ZAP-S with Bip and ER chaperones. This reviewer imagines that ZAP-S is a cytosolic protein and is hard to interact with these proteins in the ER lumen. This data should be considered as the immunoprecipitants of samples contain ER itself. The authors should clarify this point in the text.
9. For eCLIP-seq, the authors should show the overview of the data (e.g., which RNAs (non-coding RNA, mRNA, etc.) were associated, which part of the mRNA (5UTR, CDS, and 3UTR) was found in the data, etc.)
10. For Figure EV5F, the authors should show a control pie chart of the expressed mRNAs in the cells used. Given that one-fourth to one-third of proteins in the genome are targeted to the ER/secretory, the high fraction of ER-related genes could be due to the nature of the cellular proteome, but not ZAP-S specificity.
11. The author should comment on ERpQC (DOI: 10.1016/j.celrep.2015.09.047) in the discussion as well.
12. In line 364, Figure EV3A should be Figure EV3B.
13. In lines 363-364, "our data indicate that AGO2 is dispensable for RAPP-mediated translational repression" may be supported by the data. The data of Figure EV3B with Figure 2D showed that AGO2 and ZAP-S are in the same pathway (that is why a non-additive effect was observed in double depletion).

Referee #4:

The manuscript submitted by Shah et al. describes that ZAP-S is a critical regulator for the mRNA stability related to RAPP. To analyze the molecular mechanism of RAPP, the authors created the model reporter protein, PPL- Δ 2L-mCherry, which is fused an imperfect SP to N-terminal of mCherry, and screened using a genome-wide CRISPR/Cas9 system by FACS. Finally the authors found ZAP-S is a key molecule, which detects aberrancy between SP and SRP association, resulting in specific degradation of their mRNAs localized in mRNA-Ribosome-NP complex. The authors also found that the cells activated UPR and ISR components when the ZAP-S would be defective, suggesting that ZAP-S plays a central role of the correct protein targeting and cellular proteostasis.

The finding that ZAP-S specifically participates in the pre-emptive mRNA degradation, encoding secretory and membrane proteins carrying aberrant SP, is quite interesting to researchers for ISR and UPR fields. Manuscript is well written, including a lot of data set. However, I have some concerns about their claims.

1. I would like to know the location of the secretory proteins harboring aberrant SP in ZAP KO cells. The authors claim that UPR was activated when PPL- Δ 2L-mCherry was overexpressed in ZAP KO cells. The activation of UPR means the overexpressed PPL- Δ 2L-mCherry accumulated in the ER. However, Δ 2L might cleave by SPP as shown in Fig.1C. Cleaved Δ 2L is a normal mCherry protein. Uncleaved Δ 2L might mislocate in the cytosol. Therefore it is hard to understand why the UPR was induced. It is important to examine the location of Δ 2L by IF, subcellular fractionation, and the size and amounts by IB.
2. In relation to the above question, the authors should confirm the UPR activation by not only the downstream ATF5 but the upstream of UPR such as phosphorylation of UPR sensors, XBP1 splicing efficiency etc., leading to the more rigorous conclusion.
3. ZAP-L has the same structure to ZAP-S, containing important functional zinc finger motif in N-terminal region(Fig.3E), but only ZAP-S interacted with 7SLRNA in SRP. Is molecular distribution or intracellular location greatly different between ZAP-S and ZAP-L? If ZAP-L is overexpressed instead of ZAP-S in the experiments in Fig.EV4A-C, can ZAP-L interact with SRPR? Is there no interaction in vitro (Fig.EV4D) ? This point should be better explained or discussed for the readers.

Minor points:

- 1 Line 168: Which is correct, 15% (in text) or 5% (in Fig.2B)?
- 2 Lines 297-298: This sentence is overstated from the results of Fig.EV5C.
- 3 Line 315 and Fig.EV5F: The readers do not follow the meaning of 26%. The authors need to clarify the interpretation of this sentence.

Point-by-point response to referee comments

First, we would like to thank the referees for the time and effort taken to provide comments on our manuscript. To address the concerns and suggestions, we have conducted an extensive series of experiments and analyses. A full response to your comments is provided below. Referees' comments are in **blue**, and our responses are in **black**.

Referee #1:

Summary

Regulation of Aberrant Protein Production (RAPP) is a quality control pathway that degrades the nascent chains with aberrant signal peptides, as well as the translating mRNA. While this pathway was first described in 2014, there are still major gaps in our understanding of how these aberrant mRNA/nascent chains are recognized and degraded. In this study, the authors used a functional genetic screen to identify zinc finger antiviral protein (ZAP) as a key component of this pathway. Overall, this study is well-designed, informative, and will be of interest for EMBO J readers. However, there are some concerns about validation of key reagents, as well as some confusion about how ZAP fits into the existing model of RAPP.

Major comments:

1. Overall- sgRNAs that are used, outside of those in the screen, should be validated either by WB or qPCR. In particular the ZAP isoform specific sgRNAs should be validated by WB.

We have confirmed the knockdown efficiencies of sgRNAs by immunoblotting or RT-qPCR, and they have been included in the revised manuscripts (Figures 2F, 3F, EV2D, EV2F, EV2G, EV3A, EV3B, and EV3F-3H). In particular, we have confirmed the depletion of ZAP isoforms by immunoblotting (see below).

2. The authors should make it clear in the results section the cell type they are using (K562) and give a brief rationale for the choice.

A rationale has been added to the Results section (page 5, line137).

3. It is a little disconcerting that the PPL Δ 4L is only minimally reduced, unlike what was reported in the original publications. The rationale proposed (complete loss of SRP interaction) is a reasonable one but the citations given (line 116) do not support this rationale, and it isn't clear why this would vary between different cell lines. Also, this hypothesis that the higher MW PPL band represents cytosolic PPL should be confirmed by ICC or fractionation.

We thank the referee for pointing this out and share the same concern regarding the PPL- Δ 4L reporter. We had previously tested the same RAPP reporters in HeLa cells (as used in the original publications) and observed results (see panels A and B below and Figures EV1A- 1B) similar to those obtained in K562 cells. A recent study reported that fusion of the PPL- Δ 4L signal peptide to insulin coding sequence resulted in a ~25% reduction in mRNA levels, in contrast to a ~75% decrease when fused to its native prolactin coding sequence (Miller *et al*, 2024, Figure 6). We therefore reason that the observed deviation from the original study may be due to differences in the context of reporter gene— mCherry in our work versus prolactin in the original study.

In addition, we had also examined the expression levels of the Δ 4L reporter driven by promoters of different strength and found that a strong, constitutive promoter (SFFV) favored accumulation of the upper species (SP*-mCherry; see panel C below) compared to a weaker, inducible promoter (Tet ON, TRE) (compare panels B and C below). Under the same experimental conditions, immunofluorescence microscopy confirmed that SP*-mCherry is diffusely localized in the cytosol (see panels D and E below). Although the reason for the inconsistent behavior of the Δ 4L signal peptide remains unclear, we focused on the PPL- Δ 2L reporter, which behaves more consistently across experimental conditions. These additional observations have been documented in Appendix Figure 1.

A more relevant publication (Karaplis *et al*, 1995) has been cited to support the notion that SP*-mCherry likely escapes the signal peptidase in the revised manuscript.

4. For all FACS figures- it would be very helpful to plot the WT and PPL Δ 2L together so that they can be directly compared.

We have added PPL-WT in Figures 3C, 3F, 3G, and 4G in the revised manuscript for comparison.

5. The use of translation efficiency in this context.

In response to comments from Referees 1 and 2, we have replaced the polysome profiling results with a more robust quantification of the reporters at protein and mRNA levels using flow cytometry and RT-qPCR, respectively (see updated Figure 1F). These measurements allow us to calculate translation efficiency (protein production per transcript) more accurately and better align with the definition of translation efficiency.

6. In 3A the blot should also be probed for PPL to see whether there is any protein production. ICC to determine whether it is in the secretory pathway or cytosolic would also be interesting.

We had already probed for PPL and EGFP (see panel A below) and they are now included in Figure 3A (see panel A below). Immunofluorescence microscopic analyses of the Δ 2L reporter in both parental and ZAP KO cells are also included in the updated Figures 1D, 1E, EV5G, and EV5H (panel B below).

7. In 3D- it is confusing that the TE of PPL Δ 2L isn't increased with ZAP knockout. My understanding of the model is that the reduced TE of PPL Δ 2L is because translation triggers degradation; therefore transcripts with ribosomes are depleted compared to those without ribosomes. If this degradation is impeded, then that ratio should be rescued. Can the authors comment on this.

We postulate that the RAPP pathway regulates gene express through two mechanisms— mRNA degradation and translational repression, analogous to miRNA-mediated gene silencing. The observation that ZAP and AGO2 are only responsible for mRNA decay but not for translational repression (Figures 1F, 1G, 3D, and EV3A-C) suggests that these two mechanisms operate independently. This finding explains why translational repression persists even when RAPP-mediated mRNA decay is abolished. We have further expanded on this point in the Discussion section and included a new model figure (Figure 6A).

8. Method of biochemical fractionation needs to be included in methods section.

The method has been added in the Methods section under “Isolation of microsomes”.

9. EV4B immunoblot needs some additional controls to demonstrate the purity of this fraction— e.g nuclear and mitochondrial markers. Would also be helpful to use a more conventional ER marker, that isn't a known interactor of ZAP-S, eg calretinin. You should also confirm this finding on imaging- ideally staining for endogenous ZAP-S, but if those antibodies are not suitable for ICC you should at least stain for your tagged exogenous ZAP-S and confirm it is in the ER by imaging.

We appreciate the suggestions. Calnexin and VDAC have been included as ER and mitochondrial markers to strengthen the claim (see panel A below and Figure 4B). We also confirmed that ZAP-S can associate with the ER under our experimental conditions by immunofluorescence microscopy (panel B below and Figures EV4D and EV4E).

10. WB in fig 5B needs loading control.

Beta-actin has been included as a loading control.

11. Overall, I find it challenging to conceptualize how ZAP fits in the existing model of RAPP. The original model postulates that signal sequences which fail to interact with SRP come into close proximity of Ago2 and are then degraded. In this model, signal sequences must have at least a weak interaction with SRP in order to be degraded by RAPP (since ZAP is actually targeted to the mutant RNA/nascent chain through SRP). However, both in the original study, as well as in this

study, knockdown of SRP promotes robust degradation. Is this process (degradation after SRP54 knockdown) ZAP dependent? I think this is a critical experiment.

We are grateful to the referee for this insightful critique. To address the comment, we performed SRP54 knockdown in ZAP KO cells and monitored reporter expression (see below and Figure EV2G). Degradation of the reporter induced by SRP54 knockdown requires ZAP, consistent with the epistatic relationship between AGO2 and ZAP (Figure EV3A-EV3C). In light of these observations, we provided a refined working model for the RAPP QC pathway (Figure 6B). In this model, the SRP complex senses the aberrant signal peptide-ribosome complexes, and ZAP engages the complex to promote its handoff to AGO2 for mRNA degradation. These mechanistic aspects will be further explored in a subsequent publication.

Minor comment

1. Chothani et al 2019 citation (in methods) is not in references.

The citation has been added.

Referee #2:

Shah et al analyzed how RAPP is regulated by ZAP-S. They first set up a reporter for RAPP in which a signal peptide was added to mCherry. A version of this with two leucines deleted led to decreased mCherry expression. The authors then used a CRISPR screen to identify proteins required for RAPP. ZAP was one of the three hits in this screen and its depletion increased reporter expression. It appears that ZAP is required for RAPP reporter RNA decay and not translational repression. The authors then show that ZAP-S depletion, but not ZAP-L, increases reporter expression. To determine how ZAP-S regulates RAPP, they use a combination of CLIP and proteomics to identify that ZAP interacts with the SRP and binds 7SL RNA. They then propose that when ZAP is not present, UPR is triggered because RAPP-mediated quality control is not functional. While the authors show that ZAP regulates RAPP in the context of signal peptides fused to reporter proteins, the mechanism for how ZAP promotes degradation of this RNA is not clear. Overall, this is an exciting paper that shows ZAP is required for RAPP but some of the conclusions need additional data to become more convincing.

We appreciate the referee's comments. The referee is correct that we have not firmly established the precise mechanism of RNA decay in the RAPP pathway. Nonetheless, we have provided several new lines of evidence supporting the role of ZAP in this process. First, we demonstrate genetic epistasis between AGO2 and ZAP (Figure EV3A-EV3C) as well as between SRP and ZAP (Figure EV2G). Second, we show by eCLIP, mass spectrometry, and *in vitro* and *in vivo* assays that ZAP interacts with the SRP, and that this binding activity is required for RAPP. Our model now positions ZAP upstream of AGO2, promoting the handoff of aberrant signal peptide-ribosome complexes to AGO2 for mRNA degradation (Figure 6B). We hope this more conservative model better reflects our current data while providing a framework for future mechanistic studies.

Major comments:

1. Figure EV3E tests whether N4BP1 or KHNYN depletion increase reporter expression similar to ZAP. While knocking down either alone has no effect, the authors did not test co-depleting them. This needs to be done to determine if they have functionally redundant activity, which has previously been shown for HIV and LNP-mRNA (Kim et al 2025 and doi: 10.1038/s41467-024-55192-z). The authors should also test whether TRIM25 depletion increases reporter mRNA abundance to determine how ZAP's role in RAPP compares to its antiviral functions. The authors make a point that ZAP contributes to RAPP independent of its antiviral activity, but without testing the role of TRIM25 and a double knockdown for N4BP1 and KHNYN, this has not been robustly tested.

This is a valid point. We have included TRIM25 knockdown as well as N4BP1/KHNYN double knockdown to strengthen the claim. Neither condition resulted in stabilization of the reporter (see Figure EV3G and 3H).

2. The authors suggest that Figure EV4B shows that ZAP-S localizes to the ER. This biochemical fractionation is not very convincing. Immunofluorescence experiments should be used to test whether ZAP-S colocalizes with the ER in these experimental conditions. It should be noted that this has not been observed in several previous studies. ZAP-L localizes to cytoplasmic membranes, including the ER, but ZAP-S is usually diffuse in the cytoplasm. Similarly, the SRP proteins highlighted in Figure 4A represent a small number of the hundreds of proteins identified in the proteomics experiments (Table EV3). This is not strong evidence that ZAP binds the SRP.

We thank the referee for this helpful comment. To strengthen the biochemical fractionation, we have now included additional ER and mitochondrial makers. Additionally, we have performed immunofluorescence microscopy to assess the subcellular localization of ZAP-S in HeLa and K562 cells (See Referee 1, major point 9). These analyses confirmed ZAP-S colocalization with the ER, supporting out biochemical fractionation results. We noted that previous studies have reported that ZAP-S is predominantly cytosolic (see page 10, line279), our results indicate that ZAP-S can associate with the ER under specific conditions.

Regarding the proteomics data, we agree that only a subset of SRP proteins was identified by the mass spec experiment. However, the interaction was further supported by co-purification of ZAP-S with microsomes (See Referee 1, major point 9) and by RNA crosslinking to 7SL RNA (Figure 4C-

4D). We have now included co-immunoprecipitation analysis showing that SRP54 co-precipitates with ZAP-S (Figure EV4C and point 3 below). Together, these independent lines of evidence support a functional interaction between ZAP-S and the SRP complex.

3. The CLIP experiments identified a ZAP binding peak on RN7SL1. To validate this, the authors performed an EMSA (Figure 4E-F). While this showed that ZAP can bind the RNA tested, there was not a negative control RNA. Therefore, this experiment does not show that ZAP specifically binds the peak identified by CLIP. Competition EMSAs, or a similar technique, should be used to show that ZAP binds this region with specificity. Overall, the specificity of ZAP for the SRP is not clear. Immunoprecipitating ZAP and testing whether its interaction with SRP components and the 7SL RNA by western blotting and RT-qPCR, respectively, would enhance confidence in these findings.

As suggested by the referee, we performed competition EMSAs by titrating cold (non-fluorescent) 7SL RNA probe to confirm the specificity of the interaction between ZAP-S and 7SL RNA. Indeed, addition of cold probe competed away bound fluorescent 7SL RNA probe (see panel A below and Figure EV4J). Moreover, we validated the interaction between ZAP-S interaction with SRP proteins by immunoblotting and this interaction was enhanced upon induction of the $\Delta 2L$ reporter (see panel B below and Figure EV4C). Together with the data from the ZAP-S RNA-binding mutant (Figures 4E-4G), these results provide strong evidence for the specificity and biological relevance of the ZAP-S 7SL RNA interaction.

Minor comments:

1. Figure 1 analyzed how the mutated signal peptide inhibits mCherry expression. The authors say that Figures 1D and 1E show that translation is inhibited but this is not clear from the data. There is at best only a very small effect in the polysome profiling experiment shown in Figure 1D. The data in Figures 1E and EV1A are normalized to the point where it is not clear what the real effect is on mCherry expression. The RNA analysis in EV1A needs to be shown as normalized counts with mCherry and EGFP shown independently. The Ribo-seq data needs to be shown clearly so the ribosome coverage across the mCherry and GFP transcripts can be compared.

We have replaced the polysome profiling results with a direct measurement of mRNA and protein levels to better reflect the definition of translation efficiency (see Referee 1 major point 5 above). We apologize for not explaining the calculation of translation inhibition more clearly. Ribosome occupancy per mRNA was calculated by normalizing ribosome footprints to RNA-seq data for the PPL reporters. Therefore, normalization to EGFP is not required for this calculation. We have clarified this methodology in the Results section (page 5). Moreover, we now provided quantification of both RNA-seq and Ribo-seq data for the reporters (see below).

2. Similar to minor comment 1, the RNA-seq data in Figure EV3A and the Ribo-seq data in Figure 3D need to be shown as normalized counts and coverage plots to visualize the data appropriately.

Similarly, we included quantification of both RNA-seq and Ribo-seq data for the PPL- $\Delta 2L$ reporter (see below and Appendix Figure 2).

3. The authors report that ZAP-S depletion increased mCherry expression, but ZAP-L depletion had little effect (Figure 3F). Figure EV3C shows specific ZAP-S knockout but I didn't see a figure showing the ZAP-L was specifically depleted. The authors need to show this.

See Referee 1 major point 1. We have included a figure showing ZAP-L is specifically depleted.

4. The RT-qPCR results in Figure 3J shows the wild-type GRN signal peptide controls. Figure 3B, 3H and EV3A do not show the wild-type PPL signal peptide control. In figure 3J, it is clear that ZAP-S knockdown leads to a partial rescue of reporter RNA abundance. For Figure 3B, 3H and

EV3A it is unclear if ZAP depletion leads to a full or partial rescue and the wild type control needs to be present to determine this.

A PPL-WT control has been added to these figures.

Referee #3:

RAPP, a mechanism of mRNA degradation/translation repression for ER-translocation defective proteins, has been proposed by earlier studies. However, the mechanistic details and triggering factors remained elusive. The authors nicely showed that ZAP-S facilitates the pathway associated with 7SL RNA in the SRP complex. This reviewer recommends that the authors address the following points before publication.

Major points

1. The authors nicely showed that ZAP-S binds to 7SL RNA by eCLIP-Seq. This reviewer wonders whether this is constitutive or inducible with the RAPP substrate presence. If the interaction is constitutive, the authors should explain why the RAPP is not always induced for all the mRNA translated on the ER.

We thank the referee for this insightful question. Throughout the study, we primarily examined the interaction between ZAP-S and the SRP in the presence of the PPL reporter. To address this point, we performed immunoprecipitation of ZAP-S to assess its association with the SRP. We found that SRP54 co-immunoprecipitation was increased in the presence of the reporter (induced by Doxycycline) (see Referee 2 major point 3 above), suggesting that ZAP-S engagement with the SRP is substrate-dependent.

2. The authors should test ZAP-S localization on the ER by microscopic methods.

See our response to Referee 1 major point 9 above.

3. In lines 316-317, the logic of " These findings suggest that many ER-targeted genes that were upregulated in ZAP KO cells may represent endogenous cellular substrates of the RAPP pathway" does not make sense. It might be possible, however, data does not support it.

Agreed. We have removed the claim.

4. Related to point 4, the experiment of Figure 5D, using the natural context of signal sequence, evokes another question: how does ZAP-S discriminate the target? Given the overexpression of the reporter, this could be considered as the overload of an unfavored substrate may lead to RAPP (in that case, another question is why this does not occur in PPL WT reporter). Although this reviewer agreed that Figure 5D is interesting, it requires a solid explanation/experiments/analysis of why these substrates are selected for RAPP.

We are thankful that the referee found the results interesting. We selected the signal peptides of CD80 and PRSS8 not only because they are up-regulated in ZAP KO cells but also because the

hydrophobic regions of their signal peptides contain disfavored amino acids (tyrosine, asparagine, and glutamine) (see Figure EV5K). The referee's comment is spot on. Overload of such disfavored substrates by overexpression may trigger the QC pathway. While our experimental setup utilizes stable integration of the PPL reporters, which produces less mRNA and proteins compared to transient transfection, it is possible that PPL-WT, if highly overexpressed, could activate RAPP. While we acknowledge that this would be a clever demonstration of our point, we felt that the overexpression of endogenous mRNAs encoding unfavored hydrophobic regions, such as CD80 and PRSS8, have made a sufficiently strong point.

To clarify the rationale for selecting the signal peptides of CD80 and PRSS8, we have added a paragraph in the Introduction (page 2) describing favored and disfavored amino acids in the hydrophobic region of signal peptides and corresponding clarifications in the Results section. In this study, we focused on the disfavored residues within these signal peptides, although other yet-undefined features likely contribute to their recognition as RAPP substrates.

Minor points

1. While the authors show that ZAP knockout evokes the integrated stress response as evidenced by RNA-Seq and Western blotting, it remains unclear whether the increased translational efficiency of stress-responsive genes, such as ATF4, is also reflected in the ribosome profiling data. Similarly, XBP1 splicing pattern could be retrieved by RNA-Seq and ribosome profiling data.

We initially analyzed the XBP1 splicing via RNA-seq and observed reduced read coverage at the splice junction (see panel A below), indicating potential activation of the IRE1 branch of the UPR. To validate this observation, we performed RT-PCR, which confirmed increased XBP1 splicing (Figure EV5C). In addition, Ribo-seq analysis revealed bypassing of uORFs and a modest increase in ribosome occupancy across the ATF5 coding sequence (panel B below and Figure EV5D), further supporting ISR activation in ZAP KO cells.

2. The authors argue that ZAP-S may play a broader role in the RAPP pathway than AGO2, based on their observation that ZAP-S, but not AGO2, regulates the stability of GRN mRNA. However, this conclusion is drawn solely from data on GRN, which represents a limited basis for such a general claim. Therefore, the strength of this assertion should be toned down.

Agreed. We have removed the claim.

3. In the CRISPR screening section, the main text states that the top 15% of cells with an increased mCherry/EGFP ratio were sorted, whereas the figure legend indicates that the top 5% were sorted. This discrepancy should be addressed, and either the text or the figure should be corrected for consistency.

The typo has been fixed.

4. Figure 1D may require a statistical test for the authors' claim.

Given that polysome profiling experiments do not reflect the definition of translation efficiency, we have replaced it with quantifications of protein and mRNA levels (see Referee 1, major point 5).

5. At line 200, the authors may cite the latest reviews for RNA silencing.

Thank you for the suggestion. We have included a latest review.

6. To further support Figure EV3B, qPCR of reporter mRNAs should be conducted upon AGO2 knockdown in both parental and ZAP KO cells.

We have included RT-qPCR results of the reporter mRNA in Figure EV3A and 3B to strengthen the conclusion.

7. ZAP-L KO cells were not well-characterized in this paper. The Western blot and others should be shown.

Although both ZAP isoforms are expressed, complementation of ZAP KO cells with ZAP-S restored RAPP activity (Figure 4G), indicating that ZAP-S is sufficient for RAPP activity. This observation suggests ZAP-L is unlikely to play a major role in the process. Moreover, we have now included validation to confirm that ZAP-L was specifically depleted (See Referee 1, major point 1). Therefore, we did not further isolate ZAP-L KO cells, as we consider the current data are sufficient to support the conclusion.

8. In Figure 4A-B and lines 245-255, the authors described the interaction of ZAP-S with Bip and ER chaperones. This reviewer imagines that ZAP-S is a cytosolic protein and is hard to interact with these proteins in the ER lumen. This data should be considered as the immunoprecipitants of samples contain ER itself. The authors should clarify this point in the text.

Agreed. The interaction could stem from ZAP-S interaction with the ER. We have clarified this in the Results section (page 10).

9. For eCLIP-seq, the authors should show the overview of the data (e.g., which RNAs (non-coding

RNA, mRNA, etc.) were associated, which part of the mRNA (5'UTR, CDS, and 3'UTR) was found in the data, etc.)

We performed additional analyses to quantify reads mapped to ZAP-S crosslink sites across the transcriptome and examined the distribution of the reads mapped to mRNAs (see below and Figure EV4F-EV4G). While this work focuses on the interaction between ZAP-S and 7SL RNA, we did not pursue a detailed characterization of other interesting potential RNA targets identified in the dataset. Nonetheless, these data will provide a valuable resource for future studies aimed at uncovering the broader RNA-binding landscape of ZAP-S.

10. For Figure EV5F, the authors should show a control pie chart of the expressed mRNAs in the cells used. Given that one-fourth to one-third of proteins in the genome are targeted to the ER/secretory, the high fraction of ER-related genes could be due to the nature of the cellular proteome, but not ZAP-S specificity.

We agree with the referee's assessment and have noted this in the Results section (page 12, line 346).

11. The author should comment on ER pQC (DOI: 10.1016/j.celrep.2015.09.047) in the discussion as well.

We have included a new paragraph to discuss the protein degradation in the Discussion section (page 15).

12. In line 364, Figure EV3A should be Figure EV3B.

This has been corrected.

13. In lines 363-364, "our data indicate that AGO2 is dispensable for RAPP-mediated translational repression" may be supported by the data. The data of Figure EV3B with Figure 2D showed that AGO2 and ZAP-S are in the same pathway (that is why a non-additive effect was observed in double depletion).

We agree with the referee and have expanded our discussion of the epistatic relationship between ZAP and AGO2 (page 13).

Referee #4:

The manuscript submitted by Shah et al. describes that ZAP-S is a critical regulator for the mRNA stability related to RAPP. To analyze the molecular mechanism of RAPP, the authors created the model reporter protein, PPL- Δ 2L-mCherry, which is fused an imperfect SP to N-terminal of mCherry, and screened using a genome-wide CRISPR/Cas9 system by FACS. Finally the authors found ZAP-S is a key molecule, which detects aberrancy between SP and SRP association, resulting in specific degradation of their mRNAs localized in mRNA-Ribosome-NP complex. The authors also found that the cells activated UPR and ISR components when the ZAP-S would be defective, suggesting that ZAP-S plays a central role of the correct protein targeting and cellular proteostasis. The finding that ZAP-S specifically participates in the pre-emptive mRNA degradation, encoding secretory and membrane proteins carrying aberrant SP, is quite interesting to researchers for ISR and UPR fields. Manuscript is well written, including a lot of data set. However, I have some concerns about their claims.

1. I would like to know the location of the secretory proteins harboring aberrant SP in ZAP KO cells. The authors claim that UPR was activated when PPL- Δ 2L-mCherry was overexpressed in ZAP KO cells. The activation of UPR means the overexpressed PPL- Δ 2L-mCherry accumulated in the ER. However, Δ 2L might cleave by SPP as shown in Fig.1C. Cleaved Δ 2L is a normal mCherry protein. Uncleaved Δ 2L might mislocate in the cytosol. Therefore it is hard to understand why the UPR was induced. It is important to examine the location of Δ 2L by IF, subcellular fractionation, and the size and amounts by IB.

We have performed immunofluorescence staining and microsome fractionation of the PPL- Δ 2L reporter in both parental and ZAP KO cells. In both conditions, the reporter showed clear co-localization with the ER (See Referee 1, major point 6). Notably, the PPL- Δ 2L reporter was markedly enriched in the microsomal fractions in ZAP KO compared to parental cells (see below and Figure EV5I), and we did not detect SP*-mCherry species.

microsomes

Given that the ER has a limited capacity to fold and process proteins, these observations lead us to speculate that the loss of RAPP may lead to an overload of protein influx into the ER, thereby resulting in UPR activation. We have expanded the Discussion to address potential mechanisms underlying the response (page 15). Although we are keen to understand UPR activation in the absence of RAPP, we defer further characterization to future work.

2. In relation to the above question, the authors should confirm the UPR activation by not only the downstream ATF5 but the upstream of UPR such as phosphorylation of UPR sensors, XBP1 splicing efficiency etc., leading to the more rigorous conclusion.

We have shown that XBP1 splicing efficiency is elevated in ZAP KO cells (Figure EV5C) and now included evidence of PERK phosphorylation. Using Phos-tag gels which effectively separate phosphorylated from unphosphorylated species, we detected modest but reproducible PERK phosphorylation (see below and Figure 5B). We hope that the referee will consider that these additional lines of evidence as supporting our conclusion.

3. ZAP-L has the same structure to ZAP-S, containing important functional zinc finger motif in N-terminal region (Fig.3E), but only ZAP-S interacted with 7SL RNA in SRP. Is molecular distribution or intracellular location greatly different between ZAP-S and ZAP-L? If ZAP-L is overexpressed instead of ZAP-S in the experiments in Fig.EV4A-C, can ZAP-L interact with SRPR? Is there no interaction in vitro (Fig.EV4D)? This point should be better explained or discussed for the readers.

Our results showed that ZAP-L depletion did not stabilize the PPL-Δ2L reporter (Figure 3F) and complementation with ZAP-S restored RAPP activity (Figure 4G), indicating ZAP-S, but not ZAP-L, is functionally involved in RAPP. While we do not yet understand the reasons for this difference, it is plausible that the two isoforms have distinct functions, similar to previous findings showing that ZAP-S, rather than ZAP-L, inhibits SARS-CoV2 programmed ribosomal frameshifting (Zimmer *et al*, 2021). As noted by Referee 2, ZAP-S is mainly localized in the cytoplasm whereas ZAP-L is prenylated, promoting its membrane association. This difference in subcellular localization has been proposed to influence the roles these isoforms during viral infection (Charron *et al*, 2013; Schwerk *et al*, 2019). It is therefore possible that membrane anchoring of ZAP-L restricts its access to the SRP, thereby limiting its involvement in RAPP. We have included this point in the Discussion section (page 14).

Minor points:

1. Line 168: Which is correct, 15% (in text) or 5% (in Fig.2B)?

It is 15%. This typo has been fixed.

2. Lines 297-298: This sentence is overstated from the results of Fig.EV5C.

We have toned down the claim accordingly.

3. Line 315 and Fig.EV5F: The readers do not follow the meaning of 26%. The authors need to clarify the interpretation of this sentence.

As suggested by Referee 3 (minor point 10), we have clarified this point and expanded on why we tested CD80 and PRSS8.

References

- Charron G, Li MMH, MacDonald MR & Hang HC (2013) Prenylome profiling reveals S-farnesylation is crucial for membrane targeting and antiviral activity of ZAP long-isoform. *Proc Natl Acad Sci U S A* 110: 11085–90
- Karaplis AC, Lim SK, Baba H, Arnold A & Kronenberg HM (1995) Inefficient membrane targeting, translocation, and proteolytic processing by signal peptidase of a mutant preproparathyroid hormone protein. *J Biol Chem* 270: 1629–35
- Miller SC, Tikhonova EB, Hernandez SM, Dufour JM & Karamyshev AL (2024) Loss of Preproinsulin Interaction with Signal Recognition Particle Activates Protein Quality Control, Decreasing mRNA Stability. *J Mol Biol* 436: 168492
- Schwerk J, Soveg FW, Ryan AP, Thomas KR, Hatfield LD, Ozarkar S, Forero A, Kell AM, Roby JA, So L, *et al* (2019) RNA-binding protein isoforms ZAP-S and ZAP-L have distinct antiviral and immune resolution functions. *Nat Immunol* 20: 1610–1620
- Zimmer MM, Kibe A, Rand U, Pekarek L, Ye L, Buck S, Smyth RP, Cicin-Sain L & Caliskan N (2021) The short isoform of the host antiviral protein ZAP acts as an inhibitor of SARS-CoV-2 programmed ribosomal frameshifting. *Nat Commun* 12: 7193

Dear Dr. Wu,

Thank you again for the submission of your revised manuscript (EMBOJ-2025-121583R) to The EMBO Journal for our consideration, and for your patience during peer review. Your manuscript has been sent back to the four original referees who had previously assessed the first version of the work, and we have now received their comments, which are appended below.

I am pleased to say that all four referees are satisfied with the revision, mention that their initially raised concerns have been satisfactorily addressed, and find the work interesting and the advance for the field significant. Referee #1, however, identifies some issues with the new data that require clarification before we can proceed with formal acceptance of the manuscript for publication in The EMBO Journal. In addition, referee #3 detected a typo, which should also be corrected.

In light of this input, I would like to invite you to submit a final revised version of your manuscript along with a detailed point-by-point response addressing the remaining concerns of the referees. Your revision will be sent back to referee #1 for a quick round of re-review.

From the editorial side, there are also a few changes we need you to make in the final version of your manuscript, before we can move forward with its formal acceptance and publication in The EMBO Journal:

- Please provide a list of up to 5 relevant keywords after the Abstract of your revised manuscript (preferably broad terms to enhance the online search engine discoverability of your article).
- All DNA/RNA high-throughput sequencing and mass spectrometry data generated in the study must be deposited in public repositories. Please make sure that all these data will be publicly available at the time of publication, and provide the databases, accession IDs, and specific and permanent URLs to the deposited datasets, in the Data availability section of your revised manuscript.
- Please rename heading "Declaration of interests" to "Disclosure and competing interests statement".
- Heading "Material and Methods" should be renamed to "Methods".
- The author contributions statement should be removed from the manuscript file. Instead, we use CRediT to specify the contributions of each author in the journal submission system. Please feel free to use the free text box to provide more detailed descriptions during submission. See also our guide to authors for more information: <https://link.springer.com/partners/embo-press/editorial-policies#Authorship>.
- Expanded View (EV) Figures should be uploaded as individual, high-resolution Figure files with their legends in the main manuscript file placed below the main Figure legends.
- Source file names, titles, legends and manuscript callouts of Tables EV1-EV3 all need to be updated to "Dataset EV#" instead; their legends should be removed from the main manuscript file and provided instead in a separate tab/sheet in each Excel file.
- The Appendix file needs to be in PDF format; its title page should contain heading "Appendix for:", followed by the manuscript's title and a Table of Contents including page numbers for all listed items; Appendix Figures should be compiled in this Appendix PDF file with their legends placed below the corresponding Figures; the nomenclature throughout the Appendix file and the main manuscript file should be "Appendix Figure Sx" and "Appendix Table Sx".
- Please note that EMBO press papers are accompanied online by:
 - A) a short (2 sentences) summary of the findings and their significance,
 - B) 2-5 short bullet points highlighting the key results, and
 - C) a synopsis image in .jpg or .png format that is exactly 550 pixels wide and 300-600 pixels high (the height is variable). Please note that all text needs to be legible at the final size.Please upload this information along with your revised manuscript (the text for A and B should be provided in a separate Word file).
- During our routine data checks, our data editors have raised a number of concerns regarding data, Figures and their legends. Some of these issues/requests appear to have already been addressed, but not entirely. Some of the points have not been addressed yet. For your convenience, I paste below all concerns again and kindly request they be completely addressed (and highlighted) in the final version of the manuscript:
 1. Information related to "n" (e.g. the sample size, whether they are biological or technical replicates, other relevant information) must be provided in the legends of Figures 5F, EV1 A, EV2 D, EV3 A, B, H; EV5 A, B, E, F.
 2. Please note that n=2 in Figures 1G, 3D, 5D. In such cases, no statistics can be calculated and shown (e.g. error bars or similar), while the individual data points must be shown in the respective Figure panels.

3. The error bars must be defined in the legends of Figures 1E, F; 3B, H, J; EV1 A, EV2 D, EV3 A, B, H; EV4 E, EV5 A, E, F; Appendix figure 1C, 2A.
4. The annotated p-values (****/****/**/*) must be defined and the exact p-values must be provided in the legends of Figures 1B, F; 3B, H, J; 5F, EV1 A, EV2 D, EV3 A, EV5 E, F as appropriate.
5. Please indicate the statistical test used for data analysis in the legends of Figures 1B; 3B, H, J; 4A, 5F, EV1 A, EV2 D; EV3 A, EV5 B, E, F; Appendix figure 2A.
6. Please note that the scale bar needs to be defined for Figures 1D, EV1 C, EV4 D, EV5 G, Appendix figure 1B.

- The manuscript sections need to be named and ordered as follows: Title page - Abstract - Keywords - Introduction - Results - Discussion - Methods - Data Availability - Acknowledgements - Disclosure and Competing Interests Statement - References - Figure Legends - main Tables (if there are any) - Expanded View Figure Legends.

Please also note that as part of the EMBO Press transparent editorial process, The EMBO Journal publishes online a Peer Review File along with each accepted manuscript. This File will be published in conjunction with your paper and will include the referee reports, your point-by-point responses and all pertinent correspondence relating to the manuscript. Please note that your Author's Checklist will also be published at the end of the Peer Review File. Please let us know in case you want to remove any data or figures from your point-by-point responses before they are published as part of the Peer Review File. Retaining unpublished data in the Peer Review File means that these count as published and that the Peer Review File would need to be referenced in future publications. Please let the editorial office know in case you want to remove any data from this file (contact@embojournal.org).

We look forward to seeing a final version of your manuscript as soon as possible. Please let us know if you have any questions and use this link to submit your revision: *Unavailable*

Best regards,

Ioannis

Referee #1:

Regulation of Aberrant Protein Production (RAPP) is a quality control pathway that degrades the nascent chains with aberrant signal peptides, as well as the translating mRNA. While this pathway was first described in 2014, there are still major gaps in our understanding of how these aberrant mRNA/nascent chains are recognized and degraded. In this study, the authors used a functional genetic screen to identify zinc finger antiviral protein (ZAP) as a key component of this pathway. Overall, this study is well-designed, informative, and will be of interest for EMBO J readers. Additionally, the authors were very responsive to reviewer concerns, and did an excellent job addressing the comments. However, the additional data in the revised manuscript raises a few questions; I advise addressing these before publication.

Major comments:

- Given your model, I am very confused by your result that ZAP is required for mRNA decay in conditions of SRP54 KD. Per your model, ZAP is recognizing these substrates through SRP -- ZAP binds SRP, particularly 7SL; if there is a signal sequence with high affinity for SRP, ZAP is displaced. However, with low affinity signal sequences ZAP remains bound; and hands off to Ago2 for RNA decay. Therefore, how can ZAP induce mRNA decay in the absence of SRP54? How is ZAP recognizing RAPP substrates in the absence of SRP? Notably, at least in previous publications, the mRNA decay induced by SRP54 KD is most pronounced in mRNAs with WT signal sequences, rather than RAPP substrates. Your experiment was done with PPLdel2L, I wonder whether this may explain your disparate result.
- All ICC images need to show DAPI.
- 1D- Need to show eGFP and mCherry in the same cell. Also, you need a nuclear stain. Finally, it would be very helpful to also show this experiment with WT PPL.
- In EV4D, you are looking to establish co-localization of ZAP-S with ER using ICC. However, the calreticulin ICC looks extremely bizarre, not at all like ER, although it looks really good in 1D. Also, since these cells are also expressing your reporter,

it would be helpful to show mCherry and GFP channels as well to demonstrate that the ZAP-S and/or calreticulin aren't just bleedthrough of your reporters.

Minor comments:

Additionally, for clarity I recommend revising the model in Figure 6. I think this figure is confusing in its current state. If you want to keep the framework of a) as an overview and b) as more detailed schematic, I recommend trying to graphically indicate that- maybe by showing B as a "zoom in" of a. But I actually think that framework isn't helpful, and what might be more helpful would be a comparison between a WT and mutant signal sequence.

Referee #2:

The authors have addressed my comments on the manuscript satisfactorily.

Referee #3:

This reviewer appreciates the authors' efforts to improve the manuscript. This reviewer supports the publication of this work. A minor suggestion could be found as follows.

Line 270: "chaperon" should be "chaperone".

Referee #4:

Most concerns have been resolved. This revised manuscript has been greatly improved and it will give a generous interest among readers in EMBO J.

Point-by-point response to referee comments

We would like to thank the referees again for their time and effort taken to review our revised manuscript. Our responses to the referee's comments are provided below. Referees' comments are shown in **blue**, and our responses are in **black**.

Referee #1:

Regulation of Aberrant Protein Production (RAPP) is a quality control pathway that degrades the nascent chains with aberrant signal peptides, as well as the translating mRNA. While this pathway was first described in 2014, there are still major gaps in our understanding of how these aberrant mRNA/nascent chains are recognized and degraded. In this study, the authors used a functional genetic screen to identify zinc finger antiviral protein (ZAP) as a key component of this pathway. Overall, this study is well-designed, informative, and will be of interest for EMBO J readers. Additionally, the authors were very responsive to reviewer concerns, and did an excellent job addressing the comments. However, the additional data in the revised manuscript raises a few questions; I advise addressing these before publication.

Major comments:

1. Given your model, I am very confused by your result that ZAP is required for mRNA decay in conditions of SRP54 KD. Per your model, ZAP is recognizing these substrates through SRP -- ZAP binds SRP, particularly 7SL; if there is a signal sequence with high affinity for SRP, ZAP is displaced. However, with low affinity signal sequences ZAP remains bound; and hands off to Ago2 for RNA decay. Therefore, how can ZAP induce mRNA decay in the absence of SRP54? How is ZAP recognizing RAPP substrates in the absence of SRP? Notably, at least in previous publications, the mRNA decay induced by SRP54 KD is most pronounced in mRNAs with WT signal sequences, rather than RAPP substrates. Your experiment was done with PPLdel2L, I wonder whether this may explain your disparate result.

We appreciate the referee's thoughtful comment and agree that, in our working model, ZAP engages RAPP substrates through its association with the SRP. Importantly, however, depletion of SRP54 does not abolish completely SRP-ribosome interactions. A previous study has shown that 7SL-containing SRP particles can still associate with ribosomes independently of SRP54, albeit in a weaker and non-productive manner (Wild et al., 2019, PMID: 30649417). We therefore hypothesize that SRP54 depletion generates a non-productive SRP state in which ZAP can remain associated with 7SL-SRP assemblies on the ribosome and promote RAPP-dependent mRNA decay. Moreover, because SRP54 knockdown is not complete knockout, residual SRP54-containing SRP complex can facilitate localization of ZAP to secretory translation sites, enabling ER-targeted mRNA to be recognized and degraded via the RAPP pathway.

Accordingly, our revised model is agnostic regarding this point (revised Figure 6 and below) and illustrates that this complex as a non-productive SRP state that triggers the RAPP pathway, without specifying its exact composition. We hope this more conservative formulation helps integrate our data with prior observations and avoid over-interpretating the SRP composition under knockdown conditions.

We further agree with the referee that prior studies reported that SRP54 depletion preferentially destabilizes mRNAs encoding proteins with WT signal peptides (Karamyshev et al., 2014). Our experiments were performed using the PPL- Δ 2L reporter, which carries a weakened signal peptide, and differences in signal peptide composition likely influence the contribution of SRP54 during decay.

Figure 6. Proposed model for RAPP-mediated translational repression and mRNA degradation Wild-type signal peptides are efficiently targeted to the ER. Mutant signal peptides engage the SRP non-productively, leading to RAPP-mediated translational repression and mRNA degradation. ZAP-S associates with a non-productive 7SL-containing SRP complex and promotes handoff of the aberrant signal peptide-ribosome complex to AGO2 for mRNA degradation.

2. All ICC images need to show DAPI.

We have included DAPI staining in Figures 1D, EV1C, and EV5G.

3. 1D- Need to show eGFP and mCherry in the same cell. Also, you need a nuclear stain. Finally, it would be very helpful to also show this experiment with WT PPL.

We have included EGFP and mCherry staining in the same cells in updated Figure 1D and PPL-WT experiment in Figure EV1C as suggested.

4. In EV4D, you are looking to establish co-localization of ZAP-S with ER using ICC. However, the calreticulin ICC looks extremely bizarre, not at all like ER, although it looks really good in 1D. Also, since these cells are also expressing your reporter, it would be helpful to show mCherry and GFP channels as well to demonstrate that the ZAP-S and/or calreticulin aren't just bleedthrough of your reporters.

We thank the reviewer for this comment. To address the concern about potential bleedthrough from the reporter signals, we repeated the experiment to include the EGFP and mCherry (PPL- Δ 2L) channels in the analysis. As shown in the revised Figure EV4D and below, ZAP-S and

calreticulin signals are spatially distinct from mCherry (PPL- Δ 2L), confirming the observed colocalization is not due to bleedthrough.

Minor comments:

Additionally, for clarity I recommend revising the model in Figure 6. I think this figure is confusing in its current state. If you want to keep the framework of a) as an overview and b) as more detailed schematic, I recommend trying to graphically indicate that- maybe by showing B as a "zoom in" of a. But I actually think that framework isn't helpful, and what might be more helpful would be a comparison between a WT and mutant signal sequence.

We have incorporated the Referee's comments in our revised model. See Major comment 1 above.

Referee #2:

The authors have addressed my comments on the manuscript satisfactorily.

Referee #3:

This reviewer appreciates the authors' efforts to improve the manuscript. This reviewer supports the publication of this work. A minor suggestion could be found as follows.

Line 270: "chaperon" should be "chaperone".

This typo has been corrected.

Referee #4:

Most concerns have been resolved. This revised manuscript has been greatly improved and it will give a generous interest among readers in EMBO J.

Dear Colin,

Congratulations on an excellent manuscript! I am glad to say that the revised version of your work has now been seen by referee #1, who acknowledges that their concerns have been appropriately addressed (comment appended below), and I am therefore pleased to inform you that your manuscript has been accepted for publication in The EMBO Journal. Thank you very much for comprehensively addressing the initially raised referee concerns and all editorial requests for corrections and changes.

Your manuscript will now be processed for publication by EMBO Press. It will be copy edited and you will receive page proofs prior to publication. Please note that you will be contacted by Springer Nature Author Services to complete licensing and payment information.

You may qualify for financial assistance for your publication charges - either via a Springer Nature fully open access agreement or an EMBO initiative. Check your eligibility: <https://link.springer.com/journal/44318/how-to-publish-with-us>

If you have any questions, please do not hesitate to contact the Editorial Office. Thank you for your contribution to The EMBO Journal. Working with you has been a pleasure.

Best regards,

Ioannis

Referee #1:

The authors have satisfactorily addressed my comments on the manuscript.

Please note that it is The EMBO Journal policy for the transcript of the editorial process (containing referee reports and your response letters) to be published as an online supplement to each paper. If you should prefer removal of any referee-only figures included in the point-by-point response(s), e.g. because they may still be used for future publication or because they have been reproduced from published work by others, please do let us know immediately via response email.

More information is available here: <https://link.springer.com/partners/embo-press/editorial-policies#Peer%20review>